# RefineX: Learning to Refine Pre-training Data at Scale from Expert-Guided Programs

## Abstract

The foundational capabilities of large language models (LLMs) are deeply influenced by the quality of their pretraining corpora. However, enhancing data quality at scale remains a significant challenge, primarily due to the trade-off between refinement effectiveness and processing efficiency. While rule-based filtering remains the dominant paradigm, it typically operates at the document level and lacks the granularity needed to refine specific content within documents. Inspired by emerging work such as ProX, we propose **REFINEX**, a novel framework for large-scale, surgical refinement of pretraining data through programmatic editing tasks. REFINEX enables efficient and fine-grained data refinement while reliably preserving the diversity and naturalness of raw text. The core strength of REFINEX lies in its ability to distill high-quality, expert-guided end-to-end refinement results into minimal edit-based deletion programs. This high-precision distillation pipeline is used to train an efficient and reliable refine model that can systematically improve every instance in the corpus at scale. We evaluate REFINEX across from-scratch pretraining at multiple model scales and find that it consistently outperforms models trained on raw, filtered, or alternatively refined data across diverse downstream tasks. On the 750M model, REFINEX yields 2.6%-7.2% average gains on lighteval tasks, and achieves comparable performance using significantly fewer training tokens. Further analysis shows that REFINEX reliably enhances text quality with both high efficiency and precision, outperforming prior approaches such as end-to-end generation and Prox-C. These results position REFINEX as a scalable, effective, and reliable solution for optimizing pretraining data in modern LLM pipelines.

## 1 Introduction

Large language models (LLMs) (Meta, 2024; Achiam et al., 2023; Anthropic, 2024; Yang et al., 2025) represent a major milestone in the development of artificial intelligence, demonstrating impressive capabilities across a wide range of tasks, including natural language understanding (Ni et al., 2025; Mei et al., 2024), question answering (Zhuang et al., 2023), complex reasoning(Wei et al., 2022; Liu et al., 2025), and agentic task planning and execution (Fan et al., 2022; Park et al., 2023). Underpinning these capabilities is the quality of the pretraining corpus, which serves as the fundamental source of both knowledge and reasoning logic (Together, 2023; Penedo et al., 2024a).

The internet offers a vast supply of pretraining data for LLMs (Ravinder et al., 2024), but much of it is noisy and unrefined, including spam, meaningless advertisements, and corrupted or incoherent text. Such low-quality content degrades data utility and increases the risk of hallucination (Huang et al., 2023; Tonmoy et al., 2024). Thus, scalable refinement of pretraining data is increasingly viewed as a critical step to push the performance limits of LLMs. In particular, refinement must meet two key criteria: (1) **Efficiency**: given the massive data volume for pretraining corpus, refinement must be scalable and low-cost; (2) **Reliability**: it must preserve valuable information and avoid introducing additional bias from models or human preferences that could distort the essence of raw data.

Satisfying both of these criteria simultaneously is highly challenging. To ensure efficiency, heuristic methods often sacrifice granularity by operating at the document level rather than refining specific content within documents. Rule-based pipelines using document filters (Rae et al., 2021; Penedo et al., 2024a; Soldaini et al., 2024) or perplexity scores (Together, 2023) retain only data meeting

rigid criteria, resulting in limited coverage and missed opportunities. Recent work (Xie et al., 2023; Wettig et al., 2024; Yu et al., 2024; Dubey et al., 2024) incorporates LLMs for filtering, but these approaches remain coarse-grained or incur high computational costs due to their generative nature. As a result, they often discard potentially valuable data and fail to improve the quality of what is retained. The emergence of ProX (Zhou et al., 2024) offered a promising direction by showing that small distilled LMs can efficiently predict refinement programs with very few tokens, which are then executed to yield refined text at low cost. However, the ProX pipeline suffers from reliability issues due to the complexity of its distillation process. The refinement programs generated from expert models often diverge from the intended behavior, making the resulting refine model unreliable.

Inspired by ProX, we introduce **REFINEX**, a novel large-scale refinement framework that builds on efficient program-based data refinement while significantly improving reliability through carefully constructed distillation data. The key limitation of ProX lies in its training directly on noisy refinement programs derived from expert outputs, which often fail to reflect high-quality refinement operations. In contrast, REFINEX first leverages expert models to generate reliable **end-to-end** refined texts. Quality evaluation experiments show that these end-to-end outputs achieve the greatest improvements in text quality. However, while these generations offer strong quality gains, they are expensive to produce and carry over-editing risks, as model-specific preferences may be imposed during generation, compromising the original data. To mitigate this, REFINEX employs a **minimum edit distance algorithm** to identify the minimal sequence of operations required to transform the original text into the expert-refined output. It filters out potentially over-aggressive edits such as insertions and replacements that may introduce model-specific preferences, and retains only high-quality deletion operations, which are encapsulated into predefined functions. Overall, REFINEX structures the construction of distillation data into two explicit stages: first performing end-to-end refinement, then generating supervision programs by comparing the refined text with the original. This separation enables the creation of cleaner and more reliable distillation data, which is then used to train a refine model capable of generating efficient and trustworthy refinement programs at scale.

To evaluate the effectiveness of REFINEX, we conduct from-scratch pretraining at multiple model scales (350M and 750M) on corpora refined by a range of baselines. These include various document-level filtering strategies (e.g., C4, Gopher, FineWeb, and their combination), as well as Prox-C, the strongest prior fine-grained, program-based refinement method. All corpora are constructed from RedPajama (Together, 2023) with a fixed 20B-token budget and are evaluated across 10 LightEval (Fourrier et al., 2023) tasks. On the 750M model, REFINEX consistently achieves the best average performance across different data settings, yielding 2.6%-7.2% gains over other baselines. Moreover, it matches or exceeds the performance of other methods using significantly fewer training tokens, demonstrating improved data efficiency. In addition to end-task performance, we conduct a detailed evaluation of refined text using DataMan (Peng et al., 2025), which shows that REFINEX delivers the highest quality improvement aside from E2E generation-while incurring lower token overhead, introducing no new content, and avoiding the risks of over-editing. These results demonstrate that REFINEX achieves both high efficiency and strong reliability, establishing it as a practical and scalable solution for pretraining data refinement.

## 2 BACKGROUND

This section summarizes the main technical paradigms for pretraining data refinement, including quality-based filtering, end-to-end refinement, and program-based refinement. We analyze their respective strengths and limitations in terms of efficiency and reliability as follows.

**Quality-Based Filtering**  Filtering-limited refinement selects pretraining data based on estimated document quality, typically using heuristic rules or LLM-assigned scores. Heuristic filtering (Rae et al., 2021; Penedo et al., 2024a; Soldaini et al., 2024) applies predefined rules-such as thresholds on URL ratio, garbled character proportion, document length, or repetition-to efficiently remove low-quality documents. However, these rules lack flexibility for instance-level variation and often discard useful content while retaining text that still needs refinement. LLM-based methods (Xie et al., 2023; Wettig et al., 2024; Yu et al., 2024) use perplexity or model-assigned quality scores but can be unstable, biased, and computationally expensive. Importantly, most filtering operates at the document or sentence level, lacking the fine-grained resolution needed to improve content at the character or sub-span level. As a result, the reliability of the "refined" text remains limited.

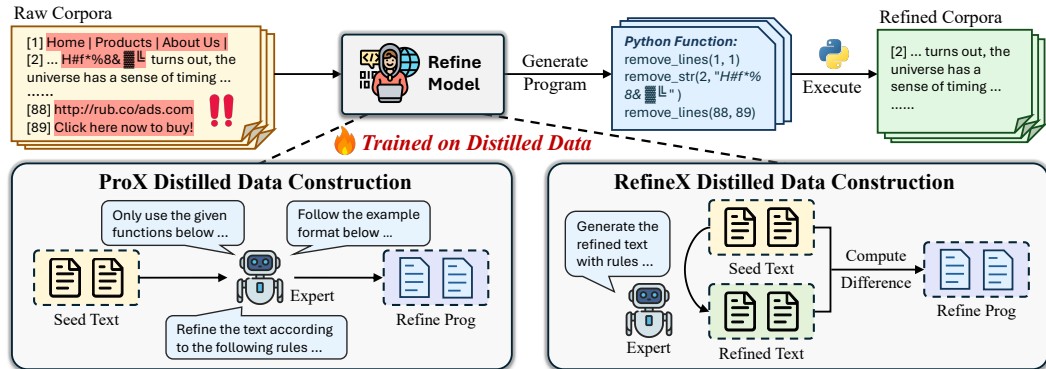

Figure 1: Overview of the program-based refinement pipeline and comparison of training data construction for the distilled refine model in ProX and REFINEX. While ProX generates refine programs using complex prompts prone to hallucination, REFINEX first produces high-quality end-to-end refined texts and derives reliable refine programs by comparing them with the original input.

**End-to-End Refinement** End-to-end refinement directly prompts a powerful LLM to rewrite a given text under refinement-specific instructions. While carefully designed prompts can yield high-quality outputs, this approach faces two key limitations in practice: (1) **It is prohibitively expensive.** Since the model generates text at the same scale as the input, end-to-end refinement incurs high inference costs, making it infeasible for large-scale deployment. (2) **It may compromise data reliability.** Despite explicitly constrained instructions, end-to-end models tend to over-edit by modifying sentence structure, over-correcting spelling, or applying stylistic preferences. These behaviors can introduce model bias and reduce the diversity of the raw data. Further implementation and performance details of this approach are discussed in Section 4.3.

**Program-Based Refinement** Program-based refinement, introduced by ProX (Zhou et al., 2024), offers a promising pipeline that jointly achieves efficiency and reliability for refinement. It prompts expert models to generate programmatic editing functions on seed examples, and trains a distilled small model to predict such refinement programs for large-scale inference and execution. The effectiveness of this approach heavily depends on the quality of the distillation data. However, prompting expert models to directly generate editing programs often fails to yield reliable supervision. This is because it requires the model not only to determine how to refine, but also to articulate the corresponding transformation logic, which introduces both modeling and supervision complexity. This often leads the trained refine model to generate hallucinated or malformed programs, resulting in execution failures or incorrect edits that compromise data quality.

In this work, we follow ProX's core pipeline by distilling a small model capable of efficiently inferring and executing refinement programs. Figure 1 illustrates the key conceptual and procedural differences between our approach and ProX. Unlike ProX, which directly prompts expert models to generate programs, REFINEX first performs end-to-end refinement and then constructs supervision programs by comparing the refined outputs with the original text. This two-stage process yields significantly more reliable supervision and effectively eliminates over-editing risks introduced during generation, ultimately leading to a more effective and robust refine model.

## 3  REFINEX: SCALABLE REFINEMENT WITH EFFICIENCY AND RELIABILITY

Figure 2 illustrates the core workflow of REFINEX. The goal of REFINEX is to reduce the difficulty for expert models to directly generate refinement programs for distillation, while preserving as many valid refinement operations from end-to-end outputs as possible. To achieve both objectives, REFINEX first prompts an expert model to generate high-quality refined text under carefully designed instructions. It then compares the refined text with the raw input to extract a reliable sequence of deletion operations based on minimal edit distance. These operations are converted into a predefined set of program functions (see Section 1), serving as trusted supervision to train a compact refine model. Once trained, the model generates reliable refinement programs through inference, which are then executed to efficiently perform fine-grained refinement across the corpus.

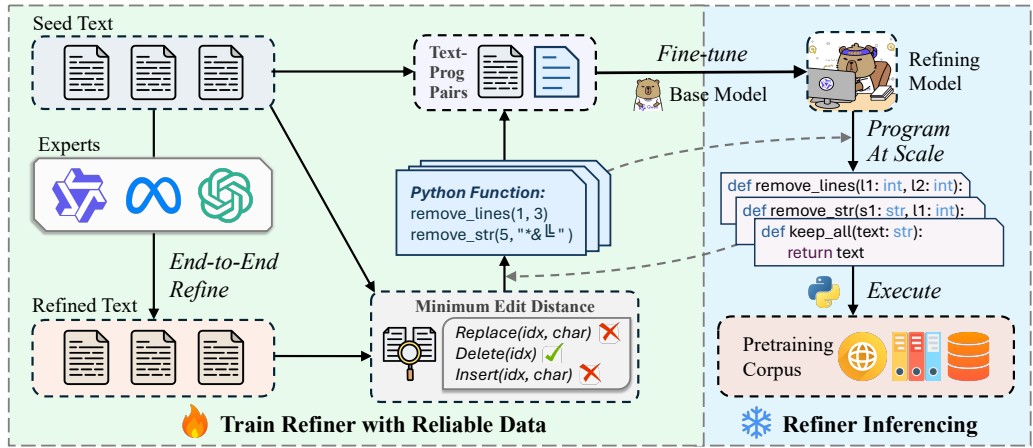

Figure 2: An overview of the REFINEX framework. (1) During training, REFINEX prompts an expert model to generate high-quality refined text under instructions, then extracts valid deletions via minimal edit distance. These are converted into program functions to supervise a reliable refine model. (2) At inference time, the trained model generates fine-grained refinement programs for each document, which are executed by a Python interpreter to produce the final refined corpus.

## 3.1 PRELIMINARIES

**Refinement Task Definition** As discussed in the comparison in Section 2, our focus is on fine-grained refinement at the character level for individual text instances, rather than document- or sentence-level quality-based filtering. Formally, given an input text $t$, we apply an executor $\mathcal{E}$ to transform it into a higher-quality version $\hat{t}$. In our REFINEX, we constrain $\mathcal{E}$ to **deletion-only operations**, allowing us to remove advertisements, meaningless URL links, random code gibberish, and other low-value content occurring anywhere in the text. Moreover, this restriction effectively prevents the introduction of model-driven stylistic preferences (Bi et al., 2025) that could compromise the authenticity of the original text-as observed in end-to-end refinement (Table 14). This also means that minor imperfections, such as spelling errors, are allowed to remain, as they are typically neutralized during large-scale pretraining through distributional memorization.

Concretely, we define a deletion operation sequence as $\mathcal{O}_{\text{del}} = \{d_1, d_2, ..., d_{|\mathcal{O}_{\text{del}}|}\}$, where each $d_j$ denotes a contiguous span of character indices to be removed. We define the following execution process to refine the original text $t$:

$$\mathcal{E}(\mathcal{O}_{\text{del}}, t) = (c_i')_{i=1}^{|t|}, \text{ where } c_i' = \text{`` if } c_i \text{ in } (d_j)_{j=1}^{|\mathcal{O}_{del}|} \text{ else } c_i \tag{1}$$

Here, $t = (c_1, c_2, ..., c_{|t|})$ denotes the original input represented as a sequence of characters. By constraining the refinement to deletion-only operations, the task formulation preserves the authenticity of the raw text while effectively removing irrelevant or low-value content.

**Program Function Design** Based on the deletion-only refinement task, we design a specific set of program functions for REFINEX. This design is crucial for achieving efficient and reliable refinement, and it must satisfy two key objectives: (1) **a minimal and simple function set** to reduce redundancy and mitigate the risk of hallucination from overly complex or ambiguous instructions, and (2) **efficient program generation** to avoid efficiency degeneration during inference. A straightforward idea is to specify exact character indices for deletion within the program. However, predicting precise character spans is extremely challenging for LLMs. Moreover, directly providing the target strings to be deleted may result in overly long deletion content, which in turn causes the generated function instructions to become excessively long in token length, significantly degrading generation efficiency.

Inspired by ProX (Zhou et al., 2024), we design three program functions in REFINEX, as summarized in Table 3.1. The `remove_lines()` function addresses long-span deletions by allowing multiple consecutive lines to be removed using a compact token representation. The `remove_str()` function supports fine-grained character-level deletion to ensure high precision. While `remove_str()` poses a risk when multiple matching substrings appear on the same line, we mitigate this through careful data selection in Section 3.2 and execution checks in Section 3.3. Additionally, the `keep_all()`

| Function Interface | Description |
|---|---|
| `remove_lines(start_line, end_line)` | Deletes all content between `start_line<int>` and `end_line<int>` |
| `remove_str(line, del_str)` | Deletes `del_str<int>` from `line<int>` only if it occurs exactly once |
| `keep_all() <str>` | Return the original text. |

Table 1: Program function definitions in REFINEX, designed to compactly cover refinement operations. Given an input text, the trained refine model outputs these functions to perform optimization.

function is introduced to indicate that no refinement is needed, avoiding empty or ambiguous outputs. These program functions are designed to be both comprehensive and minimal, reducing the risk of hallucination and maximizing refinement coverage under strict token budget constraints.

## 3.2 TRAINING A REFINER WITH RELIABLE PROGRAM SUPERVISION

The key to training a small-scale language model to generate refinement programs lies in obtaining high-quality distillation data. However, directly prompting expert LLMs to generate standard-conforming programmatic data is challenging. As noted by Zhuo et al. (2024), generating customized API calls is difficult even for state-of-the-art models, as it requires both reasoning about solution steps and formatting them into precise function calls. Although ProX improves this process using few-shot prompts and post-inference filtering, reliance on heuristics and fixed thresholds introduces significant reliability issues into the distilled data. To address this, REFINEX adopts a two-stage approach: it first generates end-to-end refined text using expert models, then derives reliable supervision programs by comparing the refined outputs with the original input. This design improves the consistency and trustworthiness of the data used to train the refine model.

**End-to-End Refinement**   Rather than directly guiding the expert model to generate refinement programs, we first instruct it to produce end-to-end refined text. Specifically, we design instruction templates aligned with the REFINEX task, which prompt the model to first reason about why a given text should be modified, and then output a refined version $t^E$. These prompts incorporate carefully constructed rules and examples, as detailed in Appendix B.2. As shown in Table 3, end-to-end refinement achieves the highest quality gains among all strategies, largely due to its token-level granularity and unconstrained rewriting flexibility. The substantial improvements over ProX are attributed to the fact that end-to-end refinement eliminates the burden of function generation, allowing the model to focus solely on improving the text.

**Function Conversion via Minimal Edit Distance**   Despite its high quality, end-to-end refinement suffers from expensive inference costs, making it impractical for large-scale deployment. Additionally, even with explicit instructions and examples, LLMs may over-edit the original text-introducing unintended stylistic changes or semantic drift (see Table 14). To reconcile reliability and efficiency, REFINEX converts the valid portions of end-to-end outputs into executable refinement programs. To achieve this, we use minimal edit distance (Levenshtein distance) (Yujian & Bo, 2007) to calculate the fewest number of edit operations required to transform the original text into the end-to-end refined version, along with the corresponding minimal operations. These edit operations include three basic types: insertion, replacement, and deletion. The principles and implementation details of the minimal edit distance algorithm can be found in Appendix B.1. Since our refinement task is limited to deletions, we capture only the deletion operations from the minimal edit distance, discarding insertions and substitutions (which typically correspond to excessive modifications in the end-to-end refinement process). The deletion operations can be formally expressed as follows:

$$\mathcal{O}_{\text{del}} \in \mathcal{O}_{\min}, \text{where } \mathcal{O}_{\min} = \{\mathcal{O}_{\text{del}}, \mathcal{O}_{\text{ins}}, \mathcal{O}_{\text{rep}}\} \text{ and } \mathcal{E}(\mathcal{O}_{\min}, t) = t^E \qquad (2)$$

Here, $\mathcal{O}_{\text{ins}}$ and $\mathcal{O}_{\text{rep}}$ denote the insertion and replacement operations under the minimal edit distance. These program representations preserve the core effects of end-to-end refinement while avoiding overcorrections, and they can be applied efficiently and reliably through token-light function calls.

**Distilling and Training the Refine Model**   To further improve the quality of supervision, we apply additional post-processing to the converted program data. We split long texts into multiple overlapping chunks to balance long-range context understanding with the limited capacity of the small-scale model. Each chunk is capped at 12k tokens, and to maximize coverage, content repetition

averagely across chunks is permitted to ensure full utilization of the model's attention window. We also enforce strict filtering criteria: transformations involving overly long insertions or substitutions, or very short deletions, are discarded to ensure reliability. Only high-confidence deletion programs are retained for distillation. To construct the training corpus, we sample large-scale seed data aligned with the distribution of real-world web-crawled corpora. We use Qwen2.5-72B-Instruct (Yang et al., 2025) as the expert model to generate end-to-end refined text, consuming approximately 12,480 GPU hours on H800-80G GPUs to process nearly 5 million raw documents. Following the conversion and filtering steps described above, we obtain a final corpus of approximately 2 million high-quality distillation examples, which are used to train a 0.6B Qwen-3-Base model as our REFINEX refiner.

### 3.3 PROGRAM EXECUTION AT SCALE

The distilled refine model can efficiently generates refinement programs for large-scale pretraining corpora using high-quality distillation data. The final refined text is obtained by sequentially executing the predefined functions according to the generated program. In rare cases where the same string appears multiple times on a single line, we skip the corresponding `remove_str()` operation to avoid unintended deletions. For long texts that are refined in multiple chunks, we apply an offset to the line indices in each chunk's program to align them with the original text, ensuring proper integration into a unified refinement. We believe that the refine model produced by the REFINEX pipeline enables scalable and reliable refinement of arbitrary text inputs.

## 4 EXPERIMENTS

### 4.1 EXPERIMENTAL SETUP

To assess the impact of refined data on model performance, we pretrain LLMs of different sizes from scratch on corpora refined by each method and evaluate them across downstream tasks (Section 4.2). We also conduct a detailed analysis of individual text instances to further investigate refinement effects (Section 4.3). The pretraining setup is detailed below.

**Pretraining Corpus and Base Models** We use RedPajama-V2 (Together, 2023), a large-scale corpus of 300 trillion tokens sourced from diverse web content, as our pretraining dataset. Starting from its publicly released 400B-token subset, we apply various refinement baselines to construct corresponding 20B-token corpora for each method. These corpora are used to pretrain two base models with 350M and 750M parameters, respectively, both following the LLAMA-2 architecture (Touvron et al., 2023b), enabling consistent evaluation across model scales.

**Evaluation Tasks and Baselines** We evaluate each pretrained model after one epoch of training on ten downstream tasks using the official implementation from LightEval (Fourrier et al., 2023). To ensure fair comparison across approaches, we consider three settings for data preparation: filtering only, fine-grained refinement only, and their combination. Fine-grained refinement is applied on top of various filtered datasets, and the full set of filtering and refinement baselines used in our experiments is described below.

- **Quality-Based Filtering:** We include several rule-based filtering baselines commonly used for corpus cleaning, including C4 (Raffel et al., 2020), Gopher (Rae et al., 2021), and FineWeb (Penedo et al., 2024a), as well as their combined variant, COMB (Go + C4 + Fw). In addition, we consider Prox-D, a state-of-the-art LLM-based filtering method that distills scoring instructions from expert LLMs to decide whether to retain each document.

- **Prox-C Refine** Prox-C directly distills the program-based refinement ability of expert LLMs and performs chunk-level refinement. We follow their setting of splitting text into 1.5K-token chunks.

Further details on the pretraining and evaluation setups can be found in Appendix C and D. In our experiments, the refine model trained with REFINEX using the Qwen3 0.6B Base (Yang et al., 2025) is applied to both raw and quality-filtered corpora. We compare its downstream performance with that of Prox C, using the respective refined datasets for pretraining.

### 4.2 EVALUATION ON PRETRAINED LANGUAGE MODELS

| Method | ARC-C | ARC-E | CSQA | HellaS | MMLU | OBQA | PIQA | SIQA | WinoG | SciQ | Avg | #Win |
|---|---|---|---|---|---|---|---|---|---|---|---|---|
| Raw | 24.5 | 45.4 | 30.5 | 38.0 | 27.5 | 28.2 | 63.8 | 39.8 | 51.0 | 67.0 | 41.6 | 0 / 10 |
| + Prox-C | 25.3 | 46.5 | 30.2 | 38.2 | 27.8 | 31.0 | 66.9 | 39.9 | 51.9 | 66.4 | 42.4 | 4 / 10 |
| + REFINEX | 25.2 | 45.9 | **32.0** | 39.4 | 27.5 | 31.2 | 66.3 | **41.0** | 52.5 | 68.3 | 42.9 | 6 / 10 |
| Rule-based filtering: GO = Gopher rules, C4 = C4 rules, FW = FineWeb rules, COMB = GO + C4 + FW. | | | | | | | | | | | | |
| GO | 24.3 | 45.1 | 28.6 | 39.7 | 27.1 | 28.4 | 66.7 | 39.4 | 50.9 | 66.9 | 41.7 | 0 / 10 |
| + Prox-C | 25.0 | 46.0 | 30.9 | 41.0 | 27.8 | 30.4 | 66.7 | 38.4 | 51.5 | 68.4 | 42.6 | 3 / 10 |
| + REFINEX | 25.0 | 48.2 | 30.4 | 41.4 | 28.1 | 29.0 | 66.7 | 40.7 | 52.8 | 69.3 | 43.2 | 7 / 10 |
| C4 | 25.3 | 44.2 | 30.9 | 39.3 | 27.4 | 29.4 | 66.5 | 39.5 | 51.3 | 67.3 | 42.1 | 0 / 10 |
| + Prox-C | 25.1 | 45.5 | 31.7 | 41.1 | 27.9 | 29.4 | 66.6 | 40.1 | 51.4 | 66.6 | 42.5 | 4 / 10 |
| + REFINEX | 24.8 | 44.5 | 31.8 | 41.3 | 28.1 | 30.4 | **68.0** | 41.2 | 50.5 | 68.2 | 42.9 | 6 / 10 |
| FW | 25.1 | 45.6 | 31.6 | 39.9 | 27.2 | 27.6 | 66.6 | 39.3 | 49.9 | 66.2 | 41.9 | 1 / 10 |
| + Prox-C | 25.5 | 44.7 | 30.8 | 39.0 | 27.9 | 29.2 | 67.4 | 39.5 | 51.3 | 67.5 | 42.2 | 3 / 10 |
| + REFINEX | 26.3 | 47.4 | 31.4 | 40.1 | 27.6 | 30.8 | 66.6 | 39.6 | 51.5 | 65.4 | 42.7 | 6 / 10 |
| COMB | 24.3 | 44.7 | 31.0 | 40.4 | 27.1 | 28.6 | 66.2 | 39.0 | **53.3** | 66.5 | 42.1 | 2 / 10 |
| + Prox-C | 24.8 | 45.4 | 31.7 | 41.1 | 27.8 | 29.4 | 66.7 | 39.5 | 51.7 | 65.8 | 42.4 | 3 / 10 |
| + REFINEX | 25.5 | 46.6 | 31.4 | **42.1** | 27.5 | 28.0 | **68.0** | 39.6 | 53.2 | 65.9 | 42.8 | 5 / 10 |
| LLM-based filtering: Prox-D | | | | | | | | | | | | |
| Prox-D | 25.1 | 45.6 | 31.6 | 39.9 | 27.2 | 27.6 | 66.6 | 39.3 | 49.9 | 67.3 | 42.0 | 1 / 10 |
| + Prox-C | 27.2 | 51.0 | 30.1 | 41.2 | 28.9 | 30.4 | 65.9 | 39.4 | 50.8 | 70.2 | 43.5 | 0 / 10 |
| + REFINEX | **28.7** | **53.2** | 30.8 | 41.7 | **29.6** | **31.8** | 67.8 | 39.9 | 51.6 | **70.9** | **44.7** | 9 / 10 |

Table 2: Performance of 750M pretrained models on 10 selected downstream tasks. All models are trained on 20B-token corpora of equal size, derived from different sources: Raw (unfiltered corpus), various filtering strategies, and further fine-grained refinement applied to the filtered corpora. Underlined results indicate the best performance within the Raw group or within each specific filtering group. **#Win** represents the number of tasks for which the method achieves the best performance within its group. **Bolded** results indicate the best overall performance across all settings.

We evaluate the effectiveness of data refinement by examining the performance of language models trained from scratch. Specifically, we begin by applying several document-level filtering methods, including Gopher, C4, FW, COMB, and Prox-D, to the raw RedPajama-V2 corpus. The filtered datasets, along with the original raw data, are then used to pretrain models from scratch, serving as our baseline setups. On top of these filtered datasets, we further apply fine-grained refinement using Prox-C and our proposed method, REFINEX, enabling a fair and comprehensive comparison across refinement strategies. For each baseline, we ensure that the post-filtering or post-refinement data contains significantly more than 20B tokens. To control for total compute, we fix the number of training steps at 10,000, with each step consisting of 1,024 global batches, and each batch containing 2048 training tokens-effectively capping the total training token budget at approximately 20B tokens. Evaluating each baseline requires training a model on its corresponding corpus, consuming approximately 1,728 GPU hours on H800-80G GPUs.

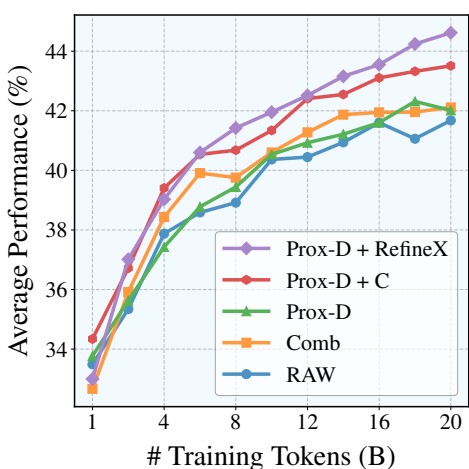

Figure 3: Downstream average performance (%) of model checkpoints with different numbers of training tokens during pretraining.

**REFINEX Excels Across Model Sizes and Downstream Tasks** We conduct pretraining using the LLAMA-2 architecture (Touvron et al., 2023b) at two model scales: 350M and 750M parameters. Evaluation is carried out using lighteval (Fourrier et al., 2023) across 10 widely-used downstream tasks covering diverse domains. As shown in Table 2, models trained with REFINEX consistently achieve the highest average scores and win the most tasks, regardless of whether refinement is applied to raw data or to previously filtered datasets. REFINEX achieves the best result on every individual task, although the top-performing variant may be derived from different data sources across tasks. When applied to data filtered by the LLM-based Prox-D method, REFINEX improves over raw data

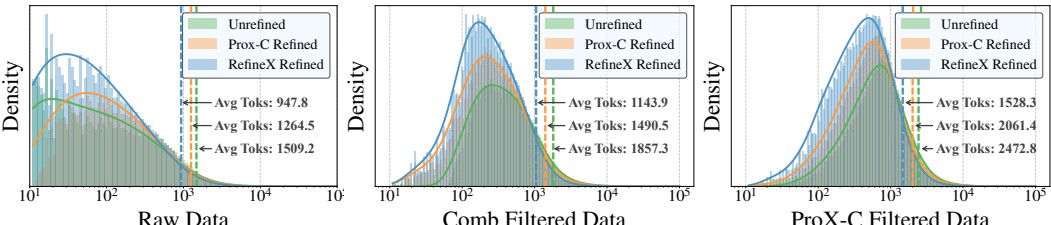

Figure 4: Token count distributions before and after refinement on raw data, rule-based filtered data (Comb), and LLM-based filtered data (Prox-C). We randomly sample 5M document instances from each corresponding baseline to ensure a representative analysis.

by +7.2%, over Comb by +5.9%, and even outperforms the strongest prior fine-grained refinement method, Prox-C, by +2.6%. Additional results for the 350M model setting are presented in Table 8.

**REFINEX Achieves Better Performance with Fewer Training Tokens** As illustrated in Figure 3, REFINEX shows clear advantages under varying training budgets. While models trained on raw or filtered data exhibit diminishing returns as token counts increase, character-level refinement methods like REFINEX continue to yield consistent gains. Notably, models trained on just 10B tokens of REFINEX-refined data match or exceed the performance of those trained on 20B tokens of Comb-filtered data, indicating that REFINEX achieves stronger performance with fewer tokens. Figure 4 further reports the token distribution after fine-grained refinement. Both rule-based and LLM-based methods help normalize the initially noisy distribution, bringing it closer to a bell-shaped curve while still showing a long-tail trend at the high-token end. Moreover, REFINEX achieves the largest overall reduction in token count while maintaining the desirable statistical structure of the data. This demonstrates that REFINEX effectively reduces per-document token cost by eliminating noisy content, thereby enabling the model to access a more diverse set of documents under the same training budget.

## 4.3 IN-DEPTH ANALYSIS OF REFINED TEXT INSTANCES

We conduct a comprehensive set of experiments to investigate how different fine-grained refinement methods affect individual text instances. To enable a more nuanced analysis across varying quality levels, we begin by pre-classifying raw text data collected from RedPajama-V2 using DataMan (Peng et al., 2025), a state-of-the-art data quality scoring tool. DataMan assigns a holistic quality score ranging from 1 to 5 to each instance, based on 14 quality dimensions and 15 application-specific signals, to estimate its pretraining utility. Specifically, we divide the data into five groups according to their scores, and randomly sample 100k instances from each group for evaluation. Based on these grouped datasets, we evaluate how each refinement method affects individual instances in terms of quality improvement, aggressive editing, refinement speed and no-change ratio.

| Quality | Method | ↑ Rat.(%) | ↓ Rat.(%) | Avg |
|---------|--------|-----------|-----------|-----|
| Score = 2 | E2E | 68.8 | 0.38 | 3.24 |
| | E2E$^{del}$ | 38.3 | 1.05 | 2.59 |
| | Prox-C | 17.2 | 0.96 | 2.23 |
| | REFINEX | 42.2 | 0.66 | 2.78 |
| Score = 3 | E2E | 59.0 | 3.39 | 3.62 |
| | E2E$^{del}$ | 33.2 | 7.95 | 3.28 |
| | Prox-C | 21.5 | 5.47 | 3.17 |
| | REFINEX | 41.2 | 4.58 | 3.45 |
| Score = 4 | E2E | 21.3 | 3.59 | 4.15 |
| | E2E$^{del}$ | 9.8 | 8.80 | 4.01 |
| | Prox-C | 8.3 | 5.64 | 4.02 |
| | REFINEX | 12.7 | 4.86 | 4.09 |

Table 3: Quality scores and changes by group after refinement. E2E$^{del}$ refers to end-to-end refinement outputs restricted to deletion-only edits.

**REFINEX Effectively Improves Text Quality** Table 3 presents the quality shift of selected groups after refinement. As expected, end-to-end (E2E) refinement achieves the highest post-refinement quality due to its costly inference and aggressive rewriting, but this also makes it impractical and risky for large-scale use. Our REFINEX ranks second in performance but operates under a far more efficient and controlled setting. It consistently improves data quality across all input groups. Compared to Prox-C, REFINEX yields greater quality improvements with a lower risk of degrading the original text. A particularly encouraging result is that REFINEX significantly outperforms its own distillation source E2E$^{del}$, used as supervision during training. This suggests that the refine model not only

learned the deletion-based programs effectively but also generalized beyond the examples it was distilled from, which highlights its potential for further development.

**REFINEX Achieves Both Efficiency and Reliability** To better observe the behavior of different methods, we conduct a detailed comparison along multiple dimensions. Figure 5 presents two complementary views. The top plot reports the ratio of output tokens to input tokens for each method, indicating their relative expansion cost. The bottom plot shows the percentage of

| Method | Score=1 | Score=2 | Score=3 | Score=4 | Score=5 |
|--------|---------|---------|---------|---------|---------|
| E2E | 5.54 | 15.06 | 16.72 | 19.26 | 11.25 |
| Prox-C | 0.32 | 0.17 | 0.08 | 0.03 | 0.01 |
| RefineX | 0.00 | 0.00 | 0.00 | 0.00 | 0.00 |

Table 4: Number of newly introduced words per 1,000 refined tokens not in the original text.

instances that remain completely unchanged after refinement, revealing the methods' tendency to preserve input content. To estimate the risk of over-editing, we compute the number of words introduced during refinement that are absent from the original text. The results are shown in Table 4.

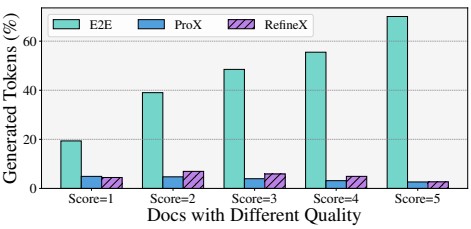

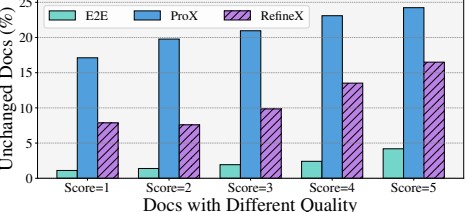

Figure 5: The top plot reflects the refinement efficiency, while the bottom plot shows the preference for leaving text untouched.

From these results, we observe that while E2E delivers the largest quality gains, it is also prohibitively slow, particularly when inference depends on large expert models. As shown in Figure 5 (top), the number of generated tokens is comparable to that of the original text, making the approach impractical at scale. Moreover, E2E suffers from excessive rewriting. This is evident from its low no-change rate in Figure 5 (bottom) and the high number of hallucinated words in Table 4, despite being explicitly prompted to avoid substantive edits and perform deletions only.

In contrast, Prox-C and REFINEX are program-based methods that achieve significantly faster refinement. This efficiency stems from the concise nature of programmatic instructions and the fast inference enabled by distilled models. According to the comparison in Figure 5, Prox-C leaves a significantly higher proportion of text untouched. However, as shown in Table 3, it still leads to more cases of quality degradation. This highlights a concerning outcome: despite making minimal changes, Prox-C worsens the quality of the text more, suggesting that its limited interventions are not only insufficient but also potentially harmful.

REFINEX applies a moderate level of edits that enhance text quality, while ensuring reliability by relying solely on deletion operations. As shown in Table 4, it introduces no additional content (i.e., zero new words), thereby avoiding risks of hallucination or over-modification. This demonstrates that REFINEX stands out by achieving both high efficiency and strong reliability. Compared to E2E, which is effective but prohibitively slow and risky, or Prox-C, which is efficient but often unreliable, REFINEX offers a principled alternative that efficiently realizes the reliability of E2E-style refinement. It achieves this through lightweight models that execute minimal and interpretable deletion-based operations. This dual advantage enables REFINEX to consistently produce refined data that is trustworthy and scalable, highlighting its potential utility in large-scale pretraining workflows.

## 5  CONCLUSION AND DISCUSSION

In this paper, we propose REFINEX, a novel approach for large-scale refinement of pretraining data. REFINEX first generates high-quality refined text in an end-to-end manner, then computes the minimal set of edit operations—based on minimum edit distance—that transform the original text into its improved version. We extract valid operations and convert them into a predefined set of programmatic functions, creating high-quality supervision data for training a compact refine model. This model is capable of inferring and executing refinement programs, enabling efficient and reliable enhancement of pretraining corpora. Extensive experiments demonstrate that REFINEX substantially improves both the quality of individual text instances and the downstream performance of models trained on the refined data. Our results highlight a promising direction for scalable and trustworthy data refinement that combines both efficiency and reliability.

## LIMITATIONS

While our experimental design comprehensively evaluates the effectiveness of REFINEX, there are two main limitations to note. First, due to resource constraints, our pretraining experiments are conducted under a limited token budget and do not cover larger-scale training regimes. Nevertheless, the consistent improvements and clear performance trends observed across model sizes and filtering setups demonstrate the strong potential of REFINEX. In particular, we simulate high-quality pretraining scenarios by applying REFINEX on top of several widely adopted filtering strategies. Second, the quality of our distillation data heavily relies on the capabilities of the expert model used during refinement. In our setup, we employ Qwen2.5-72B-Instruct, requiring approximately 12,480 GPU hours on H800s to refine 5 million raw documents. Although this model delivers competitive results, it still lags behind state-of-the-art proprietary models. In preliminary comparisons, models like GPT-4o exhibit stronger end-to-end refinement quality with fewer over-edits. As such, we believe that REFINEX has the potential to achieve even higher quality and generalization when paired with stronger expert models, assuming sufficient computational resources are available.

## ETHICS STATEMENT

This research does not involve human subjects, personal information, or sensitive data. All experiments are conducted on publicly available corpora such as RedPajama-V2 and widely used benchmark datasets. We strictly comply with dataset licenses and usage terms. The REFINEX framework is designed for academic research on efficient and reliable pretraining data refinement and does not raise additional concerns regarding fairness, privacy, or security.

## REPRODUCIBILITY STATEMENT

We have taken multiple steps to ensure reproducibility. Detailed descriptions of the datasets, preprocessing procedures, and training settings are provided in the main text (Sections 3–4) and Appendices B–D. Hyperparameters, program function definitions, and implementation details are thoroughly included in the appendix for clarity. Together, these materials enable other researchers to reliably replicate our results and further build upon our work.

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

# A  RELATED WORK

**Pretraining Data Crawl and Processing**    The quality of pretraining data is critical for shaping large language models (LLMs). Since raw web data—often from Common Crawl—is noisy and inconsistent, most LLM pipelines apply extensive preprocessing (Touvron et al., 2023a; Penedo et al., 2024a; Together, 2023). Recent work like Craw4LLM (Yu et al., 2025) and Nemotron-CC (Su et al., 2024) improves these snapshots via automated filtering, deduplication, and metadata-aware extraction. Baack (2024) critically assess Common Crawl's suitability, citing spam, duplication, and content imbalance. Pipelines typically start with document-level filtering (e.g., URL removal, boilerplate stripping, language detection)(Smith et al., 2022), followed by heuristics on length, symbol ratios, or abnormal characters(Zhang et al., 2024a; Dou et al., 2024). Deduplication uses MinHash (Broder, 1997) or n-gram hashing (Penedo et al., 2024c). While effective for filtering low-quality content, such rule-based methods are coarse and miss local noise—leading to over-filtering or incomplete cleanup.

**Filtering and Selection Strategies**    In addition to preprocessing, data filtering and selection techniques have been widely adopted to improve corpus quality. While traditional filtering approaches rely on heuristics, recent work has explored using perplexity-based scoring or LLM-judged quality signals to rank and retain high-quality documents (Xie et al., 2023; Wettig et al., 2024; Yu et al., 2024). These strategies have been extended from supervised fine-tuning into the pretraining stage, as seen in SlimPajama (Soboleva et al., 2023) and MATES (Yu et al., 2024), where learned models or proxy scoring functions resample or dynamically filter the training data. However, these methods primarily focus on selecting which documents to include or exclude, rather than improving the internal quality of documents that are partially useful but noisy. The limitations are especially evident at the sentence or token level, where heuristic filters fail to adapt and LLM scorers introduce inconsistency and bias.

**LLMs for Code Generation**    Recent advances show that large language models (LLMs) possess strong code generation and structured reasoning capabilities, making them well-suited for tasks involving executable logic (Chen et al., 2021; Austin et al., 2021; Fried et al., 2022; Wang et al., 2023). These abilities have inspired methods where LLMs produce interpretable instruction programs instead of directly generating final outputs. For example, Schick et al. (2024) and Yao et al. (2023) design prompting strategies in which LLMs emit tool-use plans or code-like sequences that are later executed. This approach reduces hallucination risks, supports modular reasoning, and enables easier error detection, as incorrect programs can be filtered or fail safely.

**Model-based Programmatic Refinement**    Motivated by these benefits, recent work explores using LLMs to generate editing programs for data refinement (Zhou et al., 2024; Zhang et al., 2023). Rather than rewriting text directly, the model produces interpretable edit operations that are applied to the original input, allowing for better quality control. However, methods like ProX (Zhou et al., 2024) prompt expert models to generate full editing programs and then distill them into smaller models. This single-step approach often yields noisy supervision and lacks fine-grained editing precision. In contrast, REFINEX follows a more structured strategy. It first prompts an expert model to produce high-quality end-to-end refined text, then derives a minimal sequence of deletion-only operations to convert the original input into the refined version. These operations are filtered for precision and distilled into a compact model, enabling interpretable, low-risk, and scalable refinement.

## B    REFINE MODEL IMPLEMENTATION IN REFINEX

### B.1    DISTILLATION DATA CONSTRUCTION AND SELECTION

| Opcode Tag | Description |
|---|---|
| equal $(i_1, i_2, j_1, j_2)$ | Span $a[i_1 : i_2]$ equals $b[j_1 : j_2]$; no change needed |
| delete $(i_1, i_2)$ | Remove span $a[i_1 : i_2]$; corresponds to deletion |
| insert $(i_1, j_1, j_2)$ | Insert $b[j_1 : j_2]$ at position $i_1$ in $a$ |
| replace $(i_1, i_2, j_1, j_2)$ | Replace $a[i_1 : i_2]$ with $b[j_1 : j_2]$ |

Table 5: Explanation of difflib.get_opcodes() edit operations used for program extraction.

**Seed Data Selection and Preprocessing**    To construct the seed dataset, we begin by scoring the raw collected corpus using DataMan (Peng et al., 2025), a state-of-the-art quality scoring model. To ensure that the seed data reflect the true distribution of the full corpus, we sample approximately 5 million documents in accordance with the score distribution. For long documents, we apply a chunking strategy by splitting them into multiple 12k-length chunks with overlapping windows. Adjacent chunks are allowed to partially overlap to ensure each chunk maintains a full 12k token span, maximizing the utility of available text.

**End-to-End Refinement with Expert LLM**    We use **Qwen2.5-72B-Instruct** (Yang et al., 2025) as the expert model to perform end-to-end refinement over the seed data. The decoding parameters are set to top-p = 0.8 and top-k = 20. We carefully design following end-to-end refinement prompts to guide the expert model in generating high-quality outputs. While over-editing cannot be entirely avoided, our prompts explicitly discourage unnecessary modifications and instead emphasize preserving valuable information. Unlike ProX, which tends to indiscriminately remove URLs or metadata, our prompts instruct the model to retain useful content, such as informative links or relevant metadata, when appropriate. Our prompt design avoids complex few-shot constructions and enables more fine-grained and reliable refinement, facilitating the creation of higher-quality distillation data.

**Filtering and Program Supervision Extraction**    We further apply data selection on the end-to-end refined text to ensure the reliability of supervision. Specifically, we discard transformations that involve overly long insertions or replacements, as well as deletions that affect only a minimal portion of the original text. To identify these operations, we compute the minimal edit operations between the original and refined text using a standard minimum edit distance algorithm, as described in Algorithm 2. In practice, we implement this using Python's difflib.get_opcodes() function, which returns a sequence of span-level operations with edit indices. The structure of these operations is summarized in Table 5. We discard any document containing insertion or replacement spans of 20 characters or more, and further filter out examples with fewer than 10 characters affected by deletions. After filtering, we obtain a set of 2 million high-quality supervision pairs. Finally, we align the resulting edit operations to our predefined set of refinement functions (Table 1) through a post-processing procedure outlined in Algorithm 1.

### B.2    MODEL TRAINING DETAILS

We fine-tune the Qwen3-0.6B-base model using the constructed high-quality supervision pairs of text and corresponding refinement programs, leveraging the LLaMA-Factory framework. In practice, we initially train multiple variants of the refine model across different model scales, including Qwen3-Base 0.6B, 1.7B, 4B, and 8B. Table 6 reports their performance in terms of text quality improvement. While the performance gap among these sizes remains relatively small, we observe that the 0.6B model provides a significant advantage in inference speed, making it the most practical choice for large-scale refinement tasks. All models are trained with a learning rate of $7 \times 10^{-6}$, a total batch size of 256, and a maximum sequence length of 16,384 tokens.

864
865
866
867
868
869
870
871
872
873
874
875
876
877
878
879
880
881
882
883
884
885
886
887
888
889
890
891
892
893
894
895
896
897
898
899
900
901
902
903
904
905
906
907
908
909
910
911
912
913
914
915
916
917

**End-to-End Refinement Prompts**

You are a careful content refiner for text relevance. Your task is to analyze the input containing a text field, and return a refined version of the text with all inter-irrelevant content removed, along with explanations of what was kept or deleted.

**Format:**
Analyze the provided input of `[doc]<text>[/doc]` and return an output in the following format:

```
modification_reason:
[doc]<brief explanation of which parts were deleted or kept, and why>[/doc]
refined_text:
[doc]<the final refined text>[/doc]
```

**Refinement Rules:**
• Delete content clearly irrelevant to the main theme or informational value, such as:
    – Website headers, navigation bars, or menu items (e.g., "Home | About | Contact")
    – Unrelated HTTP links (e.g., ads, trackers)
    – Generic footers with contact info, privacy policies, or unsubscribe links
    – Empty lines or visual dividers (e.g., "—")
• Preserve content that is informative, on-topic, or provides contextual value.
• If a sentence mixes relevant and irrelevant fragments:
    – Delete only the irrelevant fragment if the remainder stays coherent.
    – Otherwise delete the whole sentence.
• Only deletions are permitted; do not rewrite, correct, or rephrase. Preserve all original spelling, punctuation, and formatting exactly.
• If deleting leaves content incoherent or devoid of value, delete the entire sentence/section.

**Examples:**
**Example 1:**
```
Input:

[doc]The development of renewable energy technologies, particularly solar photovoltaic systems,
has reduced carbon emissions in urban areas. Recent studies show a 22% decrease in CO2 levels.[/doc]

Output:

modification_reason:
[doc]The text is highly relevant and coherent. All content discusses the impact of renewable energy
technologies on carbon emissions. Nothing was removed.[/doc]
refined_text:
[doc]The development of renewable energy technologies, particularly solar photovoltaic systems,
has reduced carbon emissions in urban areas. Recent studies show a 22% decrease in CO2 levels.[/doc]
```

**Example 2:**
```
Input:

[doc]Home | Products | About Us | Contact\nWelcome to our website\nHere's our main content...[/doc]

Output:

modification_reason:
[doc]The first line is a standard website navigation header and was removed. The other lines contain
core content and are preserved.[/doc]
refined_text:
[doc]Welcome to our website\nHere's our main content...[/doc]
```

**Example 3:**
```
Input:

[doc]Join the conversation here: http://random-forum-link.com[/doc]

Output:

modification_reason:
[doc]The only sentence consists of an unrelated URL. Removing the link would leave no meaningful
content, so the entire line was deleted.[/doc]
refined_text:
[doc][/doc]
```

**Example 4:**
```
Input:

[doc]For more information on AI governance, visit https://www.credible-ai.org/resources[/doc]

Output:

modification_reason:
[doc]Although the sentence contains a link, it is directly related to the topic of AI governance
and provides supplementary value. Therefore, it was retained in full.[/doc]
refined_text:
[doc]For more information on AI governance, visit https://www.credible-ai.org/resources[/doc]
```

*[Omitting several meaningful examples ...]*

**ACTUAL TASK EXECUTION:**
```
Input:

[doc]{input_text_task}[/doc]

Output:
```

**Algorithm 1** Mapping Valid Delete Operations to Program Functions

---

**Require:** Original text $T$ as a list of lines; Valid delete opcodes $\mathcal{D} = \{(i_1, i_2)\}$
**Ensure:** List of program functions $\mathcal{F}$
1: Split $T$ into lines $\{L_1, L_2, \ldots, L_n\}$
2: Compute character span $(s_i, e_i)$ for each line $L_i$
3: Initialize $\mathcal{F} \leftarrow [\,]$, group $\leftarrow [\,]$
4: **for** each $(i_1, i_2)$ in $\mathcal{D}$ **do**
5:     Initialize *affected_lines* $\leftarrow [\,]$
6:     **for** each line $L_\ell$ with span $[s_\ell, e_\ell]$ **do**
7:         **if** $[s_\ell, e_\ell]$ overlaps $[i_1, i_2]$ **then**
8:             **if** $i_1 \leq s_\ell$ and $i_2 \geq e_\ell$ **then**
9:                 Mark $L_\ell$ as `full-line`
10:             **else**
11:                 Extract substring $T[i_1 : i_2]$ as `del_str`
12:                 $\mathcal{F} \leftarrow \mathcal{F} \cup$ `remove_str(line=`$\ell$`, del_str)`
13:             **end if**
14:         **end if**
15:     **end for**
16:     Group consecutive `full-line` deletions into ranges
17:     **for** each range $[\ell_s, \ell_e]$ **do**
18:         $\mathcal{F} \leftarrow \mathcal{F} \cup$ `remove_lines(`$\ell_s$`, `$\ell_e$`)`
19:     **end for**
20: **end for**
21: **return** $\mathcal{F}$

---

| Quality | Model | Avg. | ↑ Rat.(%) | ↓ Rat.(%) | Untouched(%) | Empty(%) |
|---|---|---|---|---|---|---|
| $Score = 1$ | Qwen3-0.6B | 2.009 | 22.23 | 0.00 | 7.87 | 68.17 |
| | Qwen3-1.7B | 2.075 | 24.14 | 0.00 | 6.28 | 66.91 |
| | Qwen3-4B | 2.209 | 22.28 | 0.00 | 4.71 | 71.31 |
| | Qwen3-8B | 2.282 | 20.82 | 0.00 | 3.45 | 74.26 |
| $Score = 2$ | Qwen3-0.6B | 2.718 | 42.14 | 0.66 | 7.59 | 19.97 |
| | Qwen3-1.7B | 2.711 | 44.26 | 0.61 | 8.06 | 15.35 |
| | Qwen3-4B | 2.744 | 43.62 | 0.58 | 7.65 | 18.48 |
| | Qwen3-8B | 2.749 | 43.61 | 0.59 | 7.27 | 19.27 |
| $Score = 3$ | Qwen3-0.6B | 3.413 | 41.14 | 4.58 | 9.85 | 6.18 |
| | Qwen3-1.7B | 3.410 | 41.40 | 4.39 | 10.10 | 4.32 |
| | Qwen3-4B | 3.420 | 41.75 | 4.39 | 10.73 | 5.25 |
| | Qwen3-8B | 3.423 | 41.88 | 4.15 | 10.41 | 5.48 |
| $Score = 4$ | Qwen3-0.6B | 4.075 | 12.70 | 4.86 | 13.52 | 0.93 |
| | Qwen3-1.7B | 4.079 | 12.86 | 4.63 | 13.16 | 0.54 |
| | Qwen3-4B | 4.088 | 13.32 | 4.30 | 14.77 | 0.60 |
| | Qwen3-8B | 4.085 | 13.20 | 4.41 | 14.18 | 0.67 |
| $Score = 5$ | Qwen3-0.6B | 4.917 | 0.00 | 8.20 | 16.50 | 0.18 |
| | Qwen3-1.7B | 4.917 | 0.00 | 8.13 | 15.85 | 0.09 |
| | Qwen3-4B | 4.923 | 0.00 | 7.62 | 17.41 | 0.10 |
| | Qwen3-8B | 4.924 | 0.00 | 7.48 | 16.70 | 0.11 |

Table 6: Refinement statistics by group documents with different size refine models. **Avg.**, ↑ **Rat.**, and ↓ **Rat.** denote the average quality scores and the percentage of documents with improved or degraded scores after refinement, respectively. **Untouched** indicates the proportion of documents that remain unchanged, while **Empty** refers to those reduced to empty content.

**Algorithm 2** Minimum Edit Distance with Operation Extraction

**Require:** Source text $S$ of length $m$, Target text $T$ of length $n$
**Ensure:** List of edit operations $\mathcal{O}$
1: Initialize $dp[0 \ldots m][0 \ldots n] \leftarrow \infty$
2: Initialize $op[0 \ldots m][0 \ldots n]$ to store backtrack operations
3: $dp[0][0] \leftarrow 0$
4: **for** $i = 0$ to $m$ **do**
5:     $dp[i][0] \leftarrow i$; $op[i][0] \leftarrow$ Delete
6: **end for**
7: **for** $j = 0$ to $n$ **do**
8:     $dp[0][j] \leftarrow j$; $op[0][j] \leftarrow$ Insert
9: **end for**
10: **for** $i = 1$ to $m$ **do**
11:     **for** $j = 1$ to $n$ **do**
12:         **if** $S[i] = T[j]$ **then**
13:             $dp[i][j] \leftarrow dp[i-1][j-1]$
14:             $op[i][j] \leftarrow$ Match
15:         **else**
16:             ▷ Try Replace
17:             **if** $dp[i-1][j-1] + 1 < dp[i][j]$ **then**
18:                 $dp[i][j] \leftarrow dp[i-1][j-1] + 1$
19:                 $op[i][j] \leftarrow$ Replace
20:             **end if**
21:         **end if**
22:         **if** $dp[i-1][j] + 1 < dp[i][j]$ **then**
23:             $dp[i][j] \leftarrow dp[i-1][j] + 1$
24:             $op[i][j] \leftarrow$ Delete
25:         **end if**
26:         **if** $dp[i][j-1] + 1 < dp[i][j]$ **then**
27:             $dp[i][j] \leftarrow dp[i][j-1] + 1$
28:             $op[i][j] \leftarrow$ Insert
29:         **end if**
30:     **end for**
31: **end for**
32: ▷ Backtrack to extract operations
33: $i \leftarrow m, j \leftarrow n, \mathcal{O} \leftarrow [\,]$
34: **while** $i > 0$ or $j > 0$ **do**
35:     $action \leftarrow op[i][j]$
36:     Append $(action, i, j)$ to front of $\mathcal{O}$
37:     **if** $action =$ Match or Replace **then**
38:         $i \leftarrow i - 1, j \leftarrow j - 1$
39:     **else if** $action =$ Delete **then**
40:         $i \leftarrow i - 1$
41:     **else if** $action =$ Insert **then**
42:         $j \leftarrow j - 1$
43:     **end if**
44: **end whilereturn** $\mathcal{O}$

### B.3 REFINEX INFERENCE AT SCALE

We follow the general processing setup introduced in ProX (Zhou et al., 2024), where each document is split into manageable chunks before refinement. To efficiently parallelize large-scale inference, we leverage the infrastructure provided by the Datatrove project (Penedo et al., 2024b), which allows us to distribute the corpus across multiple workers-typically one per GPU, as small models do not require tensor parallelism. All refinement operations are performed using the vLLM engine (Kwon et al., 2023), which offers high-throughput execution.

For chunk-level refinement, each document is segmented into multiple overlapping spans to fit within the model's context length. While ProX restricts each chunk to roughly 1,500 tokens, our REFINEX increases the chunking window to approximately 12,000 characters, enabling the model to leverage broader semantic context across longer documents. To reduce preprocessing overhead, we approximate token-based length constraints using word counts.

The overall procedure for chunk generation is outlined below:

---

**Algorithm 3** Document Chunk Splitting Algorithm

---

**Require:** Document $D$, context window size $W$
**Ensure:** Set of chunks $C$
1:   $C \leftarrow \emptyset, c \leftarrow \emptyset$
2: **for** each line $l$ in $D$ **do**
3:     **if** TokenCount$(c + l) \leq W$ **then**
4:       $c \leftarrow c + l$                                               ▷ Add line to current chunk
5:     **else**
6:       **if** $c \neq \emptyset$ **then**
7:         $C \leftarrow C \cup \{c\}$                                  ▷ Save current chunk
8:       **end if**
9:       **if** TokenCount$(l) \leq W$ **then**
10:      $c \leftarrow l$                                               ▷ Start new chunk
11:      **else**
12:        $C \leftarrow C \cup \{\text{FlagAsSkipped}(l)\}$             ▷ Flag long line
13:        $c \leftarrow \emptyset$
14:      **end if**
15:    **end if**
16: **end for**
17: **if** $c \neq \emptyset$ **then**
18:    $C \leftarrow C \cup \{c\}$                                       ▷ Add the final chunk
19: **end if**
20: **return** $C$

---

## C PRETRAINING DETAILS

To ensure consistency with prior work, we align our pretraining pipeline with the settings adopted in ProX (Zhou et al., 2024). All models are trained using the open-source frameworks LIT-GPT(AI, 2023) and TINYLLAMA(Zhang et al., 2024b), which offer high flexibility for architecture customization and distributed training. In particular, we incorporate fused CUDA kernels from FlashAttention-2 (Dao, 2024)-including optimized rotary positional embeddings (RoPE), fused layer normalization, and efficient cross-entropy computation-to reduce memory usage during training. For scalability, we employ Fully Sharded Data Parallelism (FSDP) (Zhao et al., 2023), enabling efficient multi-node training across different model scales.

Given limited computational resources and the goal of fair comparison, we do not pretrain on the entire corpus exhaustively. Instead, we construct our training pools by randomly sampling subsets from large datasets. Specifically, for the RedPajama dataset (Together, 2023), we randomly select 70 data shards following ProX, totaling approximately 500B tokens. These are further divided into 8 equal partitions (62.5B tokens each). Since both quality-based filtering and fine-grained refinement substantially reduce the usable token count-and to varying degrees across methods-we apply refinement sequentially over the shards until the final effective token count reaches 20B.

Table 7: Pretrained model architecture and training hyper-parameters.

| Model | Hidden Size | Intermediate Size | Context Len | Heads | Layers | Vocab Size | # Params (w/o embed) |
|-------|-------------|-------------------|-------------|-------|--------|------------|----------------------|
| 350M | 1,280 | 2,048 | 2,048 | 16 | 24 | 32,000 | 354,284,800 (313,324,800) |
| 750M | 1,536 | 4,864 | 2,048 | 24 | 24 | 32,000 | 758,982,144 (709,830,144) |

| Model | Context Length | Batch Size | Max Steps | Warmup Steps | Weight Decay | Optimizer | LR Scheduler | LR |
|-------|----------------|------------|-----------|--------------|--------------|-----------|--------------|-----|
| 350M | 1,024 | 2,048 | 12,500 | 500 | 0.1 | AdamW | cosine | 5e-4 $\rightarrow$ 5e-5 |
| 750M | 1,024 | 2,048 | 12,500 | 500 | 0.1 | AdamW | cosine | 5e-4 $\rightarrow$ 5e-6 |

For all from-scratch pretraining experiments, we follow a uniform setup across model sizes. We use a learning rate of 5e-4 for both 350M and 750M models, a maximum sequence length of 2048 tokens, and a global batch size of 2 million tokens. Learning rates are scheduled using cosine decay, and additional training hyperparameters follow the configurations introduced in Zhang et al. (2024b) and Lin et al. (2024). Detailed model architecture and optimizer settings are summarized in Table 7.

## D   EXPERIMENTAL EVALUATION DETAILS

**Lighteval Benchmarks and Setup**    To assess the downstream capabilities of our pretrained models, we follow ProX to adopt a diverse set of evaluation tasks drawn primarily from the nine "early signal" benchmarks introduced in FineWeb (Penedo et al., 2024a). Evaluations are conducted using the official implementation of LIGHTEVAL(Fourrier et al., 2023), ensuring consistent and reproducible results. In addition to these nine tasks, we also include SciQ(Welbl et al., 2017) as a tenth benchmark, which has been widely adopted in recent works (Mehta et al., 2024; Wettig et al., 2024) and shown to be an informative proxy for broader model capabilities.

The complete list of evaluated datasets includes ARC-Easy and ARC-Challenge (Clark et al., 2018), CommonSenseQA (Talmor et al., 2019), HellaSwag (Zellers et al., 2019), MMLU (Hendrycks et al., 2021), OpenBookQA (Mihaylov et al., 2018), PIQA (Bisk et al., 2020), SocialIQA (Sap et al., 2019), WinoGrande (Sakaguchi et al., 2021), and SciQ. We report normalized zero-shot accuracy as the default evaluation metric across all benchmarks.

**Sampling and Aggregation Strategy**    Following the default behavior of LIGHTEVAL, we randomly sample 1,000 examples from each dataset. For MMLU, which contains 57 sub-tasks, we sample 1,000 examples from each sub-task independently and aggregate the scores into a single overall MMLU result. The final reported average is computed over all nine benchmarks, treating ARC-E and ARC-C as separate tasks while considering MMLU as a single benchmark entry.

Unlike the aggregation strategy in FineWeb, which averages across all individual subcomponents of MMLU, we adopt an equal-weighted aggregation across the nine benchmarks. This decision is motivated by the relatively high variance observed in MMLU scores and our desire to avoid overemphasizing a single benchmark in the overall evaluation. For a complete view of benchmark-specific performance and score dynamics, we provide the full set of LIGHTEVAL results in Appendix E, along with visualizations that trace model performance as a function of training tokens.

**Refined Text Evaluation Setup**    To enable a more granular analysis across varying data quality levels, we additionally evaluate the impact of refinement at the level of individual text instances. We begin by pre-classifying raw documents from the RedPajama-V2 corpus using DataMan (Peng et al., 2025), a state-of-the-art data quality scoring framework. DataMan assigns each document a holistic score from 1 to 5 by integrating signals from 14 quality dimensions and 15 application-specific heuristics, aiming to estimate its utility for pretraining. Based on these quality scores, we divide the data into five discrete groups and randomly sample 100,000 instances from each group for evaluation. For each method, we assess its refinement impact across multiple dimensions, including the post-refinement quality shift, the ratio of output to input tokens, the percentage of instances left untouched, and the number of newly introduced words that do not appear in the original text. These metrics collectively capture both the effectiveness and reliability of the refinement process.

| Method | ARC-C | ARC-E | CSQA | HellaS | MMLU | OBQA | PIQA | SIQA | WinoG | SciQ | Avg | #Win |
|---|---|---|---|---|---|---|---|---|---|---|---|---|
| Raw | 24.2 | 39.8 | 28.7 | 34.6 | 26.6 | 27.2 | 64.3 | 38.6 | 49.5 | 62.4 | 39.6 | 0 / 10 |
| + Prox-C | 24.6 | 43.9 | 29.2 | 35.5 | 27.2 | 27.2 | 64.5 | 38.8 | 50.1 | 65.4 | 40.6 | 5 / 10 |
| + REFINEX | 24.5 | 44.1 | 29.4 | 35.1 | 27.0 | 30.4 | 64.2 | 39.1 | 50.8 | 65.2 | 41.0 | 5 / 10 |
| Rule-based filtering: Go = Gopher rules, C4 = C4 rules, Fw = FineWeb rules, COMB = Go + C4 + Fw. | | | | | | | | | | | | |
| Go | 23.2 | 40.9 | 28.7 | 34.6 | 26.7 | 28.6 | 64.2 | 39.5 | 49.7 | 66.1 | 40.2 | 3 / 10 |
| + Prox-C | 23.4 | 43.1 | 28.5 | 36.5 | 26.7 | 26.4 | 64.8 | 38.4 | 51.6 | 65.7 | 40.5 | 2 / 10 |
| + REFINEX | 24.5 | 43.6 | 30.4 | 35.4 | 27.0 | 28.4 | 66.0 | 39.5 | 50.7 | 65.9 | 41.1 | 5 / 10 |
| C4 | 23.2 | 42.2 | 29.8 | 35.4 | 26.3 | 25.6 | 64.5 | 39.3 | 49.1 | 63.7 | 39.9 | 1 / 10 |
| + Prox-C | 23.4 | 43.1 | 28.5 | 36.5 | 26.7 | 26.4 | 64.8 | 38.4 | 51.6 | 65.7 | 40.5 | 2 / 10 |
| + REFINEX | 24.5 | 43.6 | 29.4 | 35.7 | 27.3 | 30.1 | 63.9 | 40.5 | 52.2 | 66.2 | 41.3 | 7 / 10 |
| Fw | 23.8 | 39.6 | 28.6 | 35.6 | 26.3 | 27.8 | 63.3 | 38.2 | 50.3 | 64.5 | 39.8 | 0 / 10 |
| + Prox-C | 24.4 | 43.2 | 28.2 | 36.0 | 27.1 | 28.2 | 64.8 | 38.8 | 51.8 | 64.9 | 40.7 | 3 / 10 |
| + REFINEX | 24.2 | 43.5 | 29.4 | 35.8 | 26.6 | 31.0 | 65.9 | 39.5 | 52.5 | 66.0 | 41.4 | 7 / 10 |
| COMB | 24.5 | 42.6 | 29.6 | 35.8 | 26.4 | 28.0 | 65.9 | 39.6 | 51.5 | 61.3 | 40.5 | 3 / 10 |
| + Prox-C | 24.9 | 42.6 | 30.0 | 36.7 | 27.1 | 27.4 | 65.1 | 38.7 | 51.0 | 64.9 | 40.8 | 3 / 10 |
| + REFINEX | 24.3 | 44.1 | 29.6 | 37.2 | 26.8 | 29.8 | 64.4 | 39.2 | 51.1 | 65.4 | 41.2 | 4 / 10 |
| LLM-based filtering: Prox-D | | | | | | | | | | | | |
| Prox-D | 25.1 | 44.3 | 28.5 | 35.6 | 27.8 | 28.8 | 64.1 | 38.5 | 51.3 | 67.2 | 41.1 | 0 / 10 |
| + Prox-C | 24.8 | 46.4 | 29.1 | 35.3 | 28.1 | 30.4 | 65.3 | 38.5 | 52.4 | 69.3 | 41.9 | 3 / 10 |
| + REFINEX | 26.8 | 48.1 | 28.6 | 36.8 | 29.3 | 29.6 | 64.4 | 39.3 | 52.8 | 69.5 | 42.5 | 7 / 10 |

Table 8: Performance of 350M pretrained models on 10 selected downstream tasks. All models are trained on 20B-token corpora of equal size, derived from different sources: Raw (unfiltered corpus), various filtering strategies, and further fine-grained refinement applied to the filtered corpora. Underlined results indicate the best performance within the Raw group or within each specific filtering group. **#Win** represents the number of tasks for which the method achieves the best performance within its group. **Bolded** results indicate the best overall performance across all settings.

# E  FULL EVALUATION RESULTS

We conduct a comprehensive set of experiments to validate the effectiveness of our proposed refinement approach under the evaluation setups introduced earlier. Our evaluation covers two main axes: (1) from-scratch pretraining at different model scales, and (2) detailed analysis of refinement effects on individual text instances.

For from-scratch pretraining, we evaluate models at both 350M and 750M parameter scales. The performance of the 350M models on 10 selected downstream tasks is summarized in Table 8. Full results across all downstream benchmarks for the 350M models are reported in Tables 10 and 11, and for the 750M models in Tables 12 and 13. These tables provide a complete breakdown of performance across different refinement and filtering strategies. In addition, we present a thorough evaluation of the quality impact of refinement on individual documents. The full set of results for this analysis is provided in Table 9, which includes multiple metrics to capture the effectiveness, efficiency, and reliability of each refinement method.

# F  CASE STUDY

To better understand the qualitative effects of different refinement methods, we present several real-world examples drawn from the RedPajama-V2 corpus. These case studies highlight key behavioral differences between end-to-end refinement, ProX, and our proposed REFINEX. In Table 14, we showcase examples of end-to-end refinement outputs. Despite carefully designed prompts intended to restrict unnecessary modifications, the expert LLM often introduces overly aggressive edits, altering the original structure or semantics of the input. In contrast, REFINEX avoids such risks by design, as it only permits deletion operations, thereby preserving the integrity of the original text and minimizing the introduction of model-specific biases.

| Quality | Method | Avg. Score | ↑ Rat.(%) | ↓ Rat.(%) | Untouched(%) | Empty(%) | #Toks |
|---|---|---|---|---|---|---|---|
| $Score = 1$ | E2E | 2.704 | 38.86 | 0.00 | 1.11 | 56.41 | 19.36 |
| | E2E$^{del}$ | 1.972 | 27.31 | 0.00 | 3.75 | 56.41 | – |
| | Prox | 1.287 | 17.05 | 0.00 | 17.12 | 33.67 | 4.91 |
| | REFINEX | 2.009 | 22.23 | 0.00 | 7.87 | 68.17 | 4.46 |
| $Score = 2$ | E2E | 3.266 | 68.83 | 0.38 | 1.39 | 12.27 | 39.02 |
| | E2E$^{del}$ | 2.590 | 38.30 | 1.05 | 6.24 | 12.27 | – |
| | Prox | 2.233 | 17.20 | 0.96 | 19.98 | 14.97 | 4.73 |
| | REFINEX | 2.718 | 42.14 | 0.66 | 7.59 | 19.97 | 6.95 |
| $Score = 3$ | E2E | 3.631 | 59.01 | 3.39 | 1.93 | 3.72 | 48.50 |
| | E2E$^{del}$ | 3.281 | 33.29 | 7.95 | 7.85 | 3.72 | – |
| | Prox | 3.192 | 21.54 | 5.47 | 21.66 | 12.80 | 3.96 |
| | REFINEX | 3.413 | 41.14 | 4.58 | 9.85 | 6.18 | 5.88 |
| $Score = 4$ | E2E | 4.175 | 21.30 | 3.59 | 2.41 | 0.26 | 55.50 |
| | E2E$^{del}$ | 4.003 | 9.85 | 8.80 | 9.92 | 0.26 | 3.16 |
| | Prox | 4.026 | 8.38 | 5.64 | 23.10 | 3.58 | 4.92 |
| | REFINEX | 4.075 | 12.70 | 4.86 | 13.52 | 0.93 | – |
| $Score = 5$ | E2E | 4.914 | 0.00 | 8.52 | 4.19 | 0.03 | 70.05 |
| | E2E$^{del}$ | 4.833 | 0.00 | 16.18 | 13.86 | 0.03 | – |
| | Prox | 4.905 | 0.00 | 9.17 | 24.24 | 0.88 | 2.64 |
| | REFINEX | 4.917 | 0.00 | 8.20 | 16.50 | 0.18 | 2.67 |

Table 9: Full evaluation results on refined documents. **Avg.**, ↑ **Rat.**, and ↓ **Rat.** represent the average quality score and the proportion of documents whose scores increased or decreased after refinement, respectively. **Untouched** indicates the percentage of documents left unchanged, and **Empty** denotes those reduced to empty content. **#Toks** indicates the ratio of output tokens to input tokens.

Beyond these examples, we present additional cases in subsequent tables to compare the refinement behaviors of REFINEX and ProX. These examples demonstrate that REFINEX is better at identifying and reliably removing low-value content-such as boilerplate text, irrelevant metadata, or malformed strings-while preserving meaningful information. The case studies further confirm that REFINEX produces more precise and trustworthy refinements, making it a more robust solution for large-scale pretraining data optimization.

Table 10: Full evaluation results (1/2) on a **350M** pretrained model across different downstream tasks, with varying numbers of training tokens used during pretraining.

| #token | ARC-C | ARC-E | CSQA | HellaSwag | MMLU | OBQA | PiQA | SIQA | WinoG | SciQ | AVG |
|---|---|---|---|---|---|---|---|---|---|---|---|
| | | | | | Raw | | | | | | |
| 4 | 22.1 | 37.3 | 25.5 | 30.4 | 25.7 | 25.8 | 61.2 | 38.1 | 50.6 | 60.3 | 37.7 |
| 8 | 23.0 | 38.6 | 27.8 | 31.0 | 26.2 | 26.8 | 62.4 | 37.4 | 49.4 | 62.7 | 38.5 |
| 12 | 23.8 | 38.3 | 27.6 | 31.7 | 26.8 | 27.6 | 62.0 | 38.9 | 49.3 | 63.8 | 39.0 |
| 16 | 23.8 | 41.0 | 28.8 | 33.8 | 26.5 | 27.4 | 64.0 | 39.4 | 49.2 | 64.3 | 39.8 |
| 20 | 24.2 | 39.8 | 28.7 | 34.6 | 26.6 | 27.2 | 64.3 | 38.6 | 49.5 | 62.4 | 39.6 |
| | | | | | Raw + ProX | | | | | | |
| 4 | 22.2 | 37.9 | 25.9 | 30.1 | 25.5 | 25.6 | 60.1 | 38.6 | 51.0 | 58.1 | 37.4 |
| 8 | 23.5 | 40.2 | 27.9 | 32.6 | 26.2 | 25.6 | 61.4 | 38.5 | 51.2 | 60.8 | 38.8 |
| 12 | 24.1 | 43.3 | 28.7 | 32.6 | 26.9 | 25.8 | 63.1 | 39.9 | 52.6 | 64.2 | 40.1 |
| 16 | 24.0 | 43.6 | 29.3 | 34.2 | 26.9 | 27.6 | 63.4 | 39.5 | 52.0 | 64.1 | 40.5 |
| 20 | 24.6 | 43.9 | 29.2 | 35.5 | 27.2 | 27.2 | 64.5 | 38.8 | 50.1 | 65.4 | 40.6 |
| | | | | | Raw + RefineX | | | | | | |
| 4 | 23.9 | 37.6 | 25.6 | 30.1 | 25.8 | 26.8 | 60.0 | 38.1 | 49.9 | 58.8 | 37.7 |
| 8 | 24.2 | 40.3 | 25.7 | 32.4 | 26.2 | 27.8 | 62.4 | 39.3 | 50.8 | 61.3 | 39.0 |
| 12 | 24.3 | 42.6 | 27.9 | 34.7 | 26.6 | 29.2 | 62.9 | 39.5 | 51.7 | 63.9 | 40.3 |
| 16 | 24.2 | 42.9 | 28.8 | 35.3 | 26.7 | 29.8 | 62.8 | 38.8 | 51.6 | 64.7 | 40.6 |
| 20 | 24.5 | 44.1 | 28.4 | 35.7 | 27.0 | 30.4 | 64.2 | 38.9 | 51.5 | 65.2 | 40.9 |
| | | | | | Go | | | | | | |
| 4 | 23.3 | 36.4 | 26.1 | 27.5 | 25.6 | 25.2 | 60.5 | 38.0 | 49.8 | 59.5 | 37.2 |
| 8 | 22.4 | 38.8 | 26.6 | 30.7 | 26.0 | 28.0 | 61.7 | 38.9 | 52.2 | 63.8 | 38.9 |
| 12 | 22.6 | 39.9 | 27.9 | 33.1 | 25.7 | 28.8 | 63.4 | 38.4 | 50.6 | 64.3 | 39.5 |
| 16 | 22.4 | 39.7 | 29.8 | 33.4 | 26.6 | 29.8 | 63.3 | 39.1 | 50.4 | 65.9 | 40.0 |
| 20 | 23.2 | 40.9 | 28.7 | 34.6 | 26.7 | 28.6 | 64.2 | 39.5 | 49.7 | 66.1 | 40.2 |
| | | | | | Go + ProX | | | | | | |
| 4 | 23.3 | 37.6 | 26.8 | 27.6 | 26.1 | 24.6 | 60.6 | 39.7 | 50.7 | 59.1 | 37.6 |
| 8 | 23.4 | 39.2 | 28.3 | 32.3 | 26.3 | 26.8 | 63.9 | 39.2 | 51.0 | 61.8 | 39.2 |
| 12 | 23.9 | 41.3 | 27.8 | 34.0 | 26.8 | 28.0 | 64.7 | 39.8 | 51.1 | 64.8 | 40.2 |
| 16 | 25.1 | 42.4 | 28.5 | 35.1 | 26.7 | 27.2 | 64.9 | 38.8 | 50.7 | 63.9 | 40.3 |
| 20 | 23.4 | 43.1 | 28.5 | 36.5 | 26.7 | 26.4 | 64.8 | 38.4 | 51.6 | 65.7 | 40.5 |
| | | | | | Go + RefineX | | | | | | |
| 4 | 20.4 | 37.4 | 27.1 | 29.6 | 25.6 | 27.2 | 60.7 | 39.0 | 50.0 | 59.6 | 37.7 |
| 8 | 22.0 | 42.0 | 29.7 | 32.3 | 25.8 | 28.2 | 62.1 | 39.6 | 51.4 | 63.2 | 39.6 |
| 12 | 23.0 | 41.7 | 28.9 | 33.6 | 26.7 | 28.4 | 64.4 | 39.5 | 51.1 | 65.3 | 40.3 |
| 16 | 23.2 | 42.1 | 29.2 | 35.2 | 26.8 | 29.0 | 65.8 | 38.4 | 50.1 | 65.9 | 40.6 |
| 20 | 24.5 | 43.6 | 30.4 | 35.4 | 27.0 | 28.4 | 66.0 | 39.5 | 50.7 | 65.9 | 41.1 |
| | | | | | C4 | | | | | | |
| 4 | 22.2 | 37.5 | 24.5 | 29.7 | 25.8 | 25.8 | 59.8 | 38.1 | 51.2 | 57.1 | 37.2 |
| 8 | 22.6 | 39.7 | 28.1 | 31.3 | 26.1 | 26.2 | 62.4 | 37.5 | 51.3 | 61.3 | 38.7 |
| 12 | 23.0 | 43.5 | 28.5 | 31.7 | 27.1 | 27.4 | 63.9 | 37.6 | 49.1 | 63.7 | 39.5 |
| 16 | 24.5 | 42.1 | 28.7 | 35.1 | 26.3 | 26.4 | 64.5 | 37.9 | 50.3 | 61.6 | 39.7 |
| 20 | 23.2 | 42.2 | 30.4 | 35.4 | 26.3 | 25.6 | 64.5 | 39.3 | 49.1 | 63.7 | 39.9 |
| | | | | | C4 + ProX | | | | | | |
| 4 | 22.7 | 37.4 | 25.4 | 28.9 | 25.9 | 27.0 | 59.4 | 37.2 | 50.2 | 58.2 | 37.2 |
| 8 | 23.5 | 39.9 | 27.9 | 32.8 | 26.2 | 26.4 | 61.1 | 38.5 | 52.0 | 65.5 | 39.4 |
| 12 | 23.1 | 41.3 | 28.5 | 34.6 | 26.4 | 27.6 | 63.4 | 38.9 | 51.4 | 65.9 | 40.1 |
| 16 | 23.2 | 42.6 | 28.0 | 34.7 | 26.7 | 27.4 | 63.6 | 40.2 | 51.6 | 65.4 | 40.3 |
| 20 | 23.4 | 43.1 | 28.5 | 36.5 | 26.7 | 26.4 | 64.8 | 38.4 | 51.6 | 65.7 | 40.5 |
| | | | | | C4 + RefineX | | | | | | |
| 4 | 22.7 | 36.9 | 25.7 | 31.2 | 26.0 | 26.2 | 60.8 | 37.1 | 51.7 | 60.0 | 37.8 |
| 8 | 22.9 | 40.9 | 28.6 | 32.7 | 26.2 | 27.6 | 62.5 | 38.8 | 54.0 | 62.6 | 39.7 |
| 12 | 23.6 | 41.4 | 28.4 | 34.3 | 26.7 | 28.9 | 62.6 | 39.0 | 52.4 | 64.6 | 40.2 |
| 16 | 23.9 | 42.6 | 29.2 | 34.8 | 27.0 | 29.3 | 62.8 | 39.8 | 52.0 | 65.0 | 40.6 |
| 20 | 24.5 | 43.6 | 29.4 | 35.7 | 27.3 | 30.1 | 63.9 | 40.5 | 52.2 | 66.2 | 41.3 |

Table 11: Full evaluation results (2/2) on a **350M** pretrained model across different downstream tasks, with varying numbers of training tokens used during pretraining.

| #token | ARC-C | ARC-E | CSQA | HellaSwag | MMLU | OBQA | PiQA | SIQA | WinoG | SciQ | AVG |
|---|---|---|---|---|---|---|---|---|---|---|---|
| | | | | | Fw | | | | | | |
| 4 | 21.5 | 34.8 | 25.1 | 29.3 | 25.4 | 24.0 | 59.4 | 38.9 | 50.8 | 58.2 | 36.7 |
| 8 | 22.8 | 39.0 | 25.5 | 30.0 | 26.0 | 26.4 | 63.1 | 38.8 | 50.7 | 60.6 | 38.3 |
| 12 | 23.2 | 39.2 | 28.1 | 32.7 | 26.0 | 26.4 | 64.3 | 39.0 | 51.6 | 63.1 | 39.4 |
| 16 | 23.8 | 41.8 | 28.7 | 35.1 | 26.3 | 27.6 | 64.3 | 39.8 | 50.3 | 65.1 | 40.3 |
| 20 | 23.8 | 39.6 | 28.9 | 35.6 | 26.3 | 27.8 | 64.5 | 38.8 | 50.8 | 64.9 | 40.1 |
| | | | | | Fw + ProX | | | | | | |
| 4 | 22.4 | 37.1 | 26.4 | 30.8 | 25.9 | 27.0 | 60.4 | 38.3 | 51.0 | 57.8 | 37.7 |
| 8 | 22.3 | 39.1 | 28.0 | 31.8 | 26.4 | 27.6 | 63.0 | 39.7 | 50.3 | 63.3 | 39.1 |
| 12 | 24.2 | 42.4 | 28.5 | 33.0 | 26.4 | 28.0 | 62.3 | 39.8 | 52.5 | 64.1 | 40.1 |
| 16 | 23.9 | 42.1 | 28.9 | 33.9 | 27.0 | 28.8 | 64.1 | 39.1 | 51.2 | 65.2 | 40.4 |
| 20 | 24.4 | 43.2 | 28.2 | 36.0 | 27.1 | 28.2 | 64.8 | 38.8 | 51.8 | 64.9 | 40.7 |
| | | | | | Fw + RefineX | | | | | | |
| 4 | 23.4 | 37.1 | 26.5 | 29.2 | 25.8 | 26.6 | 59.9 | 38.3 | 50.5 | 60.4 | 37.8 |
| 8 | 24.2 | 40.5 | 29.0 | 31.4 | 26.4 | 26.8 | 61.4 | 38.2 | 52.7 | 61.8 | 39.2 |
| 12 | 24.0 | 40.9 | 28.7 | 33.0 | 26.3 | 28.8 | 63.5 | 39.4 | 52.9 | 64.1 | 40.2 |
| 16 | 22.4 | 42.5 | 29.2 | 35.5 | 26.3 | 28.8 | 63.6 | 38.5 | 51.9 | 66.2 | 40.5 |
| 20 | 24.2 | 43.5 | 29.4 | 35.8 | 26.6 | 31.0 | 65.9 | 39.5 | 52.5 | 66.0 | 41.4 |
| | | | | | Comb | | | | | | |
| 4 | 22.1 | 36.5 | 25.7 | 29.9 | 25.5 | 26.2 | 62.5 | 38.7 | 48.7 | 56.0 | 37.2 |
| 8 | 22.8 | 40.2 | 28.6 | 32.2 | 25.9 | 26.4 | 62.1 | 39.2 | 48.8 | 60.5 | 38.7 |
| 12 | 23.5 | 40.6 | 29.2 | 33.2 | 26.2 | 27.6 | 64.9 | 38.6 | 51.0 | 64.5 | 39.9 |
| 16 | 23.1 | 41.4 | 28.8 | 35.1 | 26.5 | 26.8 | 64.0 | 40.2 | 52.0 | 62.1 | 40.0 |
| 20 | 24.5 | 42.6 | 29.6 | 35.8 | 26.4 | 28.0 | 65.9 | 39.6 | 51.5 | 61.3 | 40.5 |
| | | | | | Comb + ProX | | | | | | |
| 4 | 22.2 | 36.8 | 26.1 | 30.4 | 25.7 | 25.4 | 60.8 | 37.9 | 50.5 | 55.1 | 37.1 |
| 8 | 23.5 | 38.9 | 28.0 | 31.8 | 26.7 | 26.4 | 61.2 | 38.1 | 50.5 | 60.8 | 38.6 |
| 12 | 23.7 | 41.1 | 28.6 | 35.3 | 26.7 | 25.6 | 63.3 | 37.9 | 51.6 | 61.1 | 39.5 |
| 16 | 24.6 | 42.7 | 30.7 | 36.3 | 26.6 | 28.0 | 63.7 | 38.4 | 52.6 | 62.6 | 40.6 |
| 20 | 24.9 | 42.6 | 30.0 | 36.7 | 27.1 | 27.4 | 65.1 | 38.7 | 51.0 | 64.9 | 40.8 |
| | | | | | Comb + RefineX | | | | | | |
| 4 | 24.2 | 39.0 | 26.2 | 28.8 | 26.0 | 26.0 | 60.2 | 38.3 | 50.4 | 56.9 | 38.0 |
| 8 | 23.9 | 40.8 | 27.8 | 31.7 | 26.1 | 26.6 | 63.3 | 39.7 | 51.8 | 62.4 | 39.4 |
| 12 | 24.2 | 41.8 | 28.6 | 34.8 | 26.3 | 27.0 | 63.5 | 38.0 | 49.7 | 63.0 | 39.7 |
| 16 | 23.9 | 43.8 | 29.4 | 36.2 | 26.7 | 30.0 | 65.2 | 39.1 | 52.4 | 63.5 | 41.0 |
| 20 | 24.3 | 44.1 | 29.6 | 37.2 | 26.8 | 29.8 | 64.4 | 39.2 | 51.1 | 65.4 | 41.2 |
| | | | | | ProX-D | | | | | | |
| 4 | 23.9 | 41.6 | 25.3 | 30.0 | 25.8 | 26.2 | 59.5 | 37.7 | 49.9 | 62.2 | 38.2 |
| 8 | 24.8 | 43.2 | 27.6 | 31.5 | 26.9 | 27.8 | 62.0 | 37.8 | 51.7 | 68.4 | 40.2 |
| 12 | 24.7 | 44.5 | 28.2 | 34.1 | 27.5 | 30.2 | 61.2 | 38.4 | 51.5 | 66.5 | 40.7 |
| 16 | 24.9 | 45.4 | 28.6 | 35.3 | 27.8 | 28.8 | 64.1 | 38.0 | 51.2 | 66.5 | 41.1 |
| 20 | 25.1 | 44.3 | 28.5 | 35.6 | 27.8 | 28.8 | 64.1 | 38.5 | 51.3 | 67.2 | 41.1 |
| | | | | | ProX-D + ProX | | | | | | |
| 4 | 23.5 | 42.6 | 26.3 | 30.0 | 25.9 | 27.3 | 59.5 | 37.8 | 49.3 | 63.3 | 38.5 |
| 8 | 22.8 | 43.2 | 28.6 | 35.7 | 26.2 | 28.9 | 62.1 | 38.1 | 49.7 | 64.5 | 40.0 |
| 12 | 24.0 | 44.7 | 28.7 | 36.3 | 26.9 | 29.2 | 65.9 | 37.6 | 51.3 | 65.7 | 41.0 |
| 16 | 25.1 | 46.4 | 29.5 | 35.2 | 27.8 | 30.2 | 64.5 | 38.4 | 51.9 | 67.8 | 41.7 |
| 20 | 24.8 | 46.4 | 29.1 | 35.3 | 28.1 | 30.4 | 65.3 | 38.5 | 52.4 | 69.3 | 42.0 |
| | | | | | ProX-D + RefineX | | | | | | |
| 4 | 23.1 | 39.7 | 25.5 | 30.8 | 26.4 | 28.4 | 59.3 | 38.4 | 50.5 | 58.5 | 38.1 |
| 8 | 23.8 | 43.8 | 28.1 | 33.4 | 27.4 | 29.4 | 62.2 | 37.8 | 49.9 | 67.9 | 40.4 |
| 12 | 25.1 | 46.6 | 27.5 | 34.8 | 27.6 | 30.1 | 63.6 | 37.9 | 51.1 | 65.2 | 41.0 |
| 16 | 25.9 | 47.5 | 27.8 | 35.3 | 27.9 | 30.5 | 64.5 | 38.8 | 51.9 | 67.7 | 41.8 |
| 20 | 26.8 | 48.1 | 28.6 | 36.8 | 29.3 | 29.6 | 64.4 | 39.3 | 52.8 | 69.5 | 42.5 |

Table 12: Full evaluation results (1/2) on a **750M** pretrained model across different downstream tasks, with varying numbers of training tokens used during pretraining.

| #token | ARC-C | ARC-E | CSQA | HellaSwag | MMLU | OBQA | PiQA | SIQA | WinoG | SciQ | AVG |
|--------|-------|-------|------|-----------|------|------|------|------|-------|------|-----|
| Raw | | | | | | | | | | | |
| 4 | 22.9 | 40.1 | 26.4 | 30.5 | 26.1 | 24.2 | 60.4 | 38.7 | 49.8 | 55.5 | 37.5 |
| 8 | 24.3 | 42.6 | 27.9 | 34.8 | 26.7 | 27.2 | 64.4 | 40.2 | 48.0 | 63.1 | 39.9 |
| 12 | 24.2 | 43.6 | 28.5 | 36.7 | 27.2 | 27.8 | 65.8 | 38.8 | 48.6 | 64.2 | 40.5 |
| 16 | 24.7 | 45.2 | 30.5 | 38.1 | 27.1 | 29.6 | 66.8 | 39.5 | 50.6 | 66.1 | 41.8 |
| 20 | 24.5 | 45.4 | 30.5 | 38.0 | 27.5 | 28.2 | 63.8 | 39.8 | 51.0 | 67.0 | 41.6 |
| Raw + ProX | | | | | | | | | | | |
| 4 | 24.1 | 38.5 | 26.8 | 30.6 | 26.2 | 25.2 | 61.6 | 38.6 | 50.7 | 56.4 | 37.9 |
| 8 | 23.5 | 40.1 | 28.4 | 33.6 | 26.5 | 27.0 | 62.4 | 37.7 | 52.1 | 57.9 | 38.9 |
| 12 | 24.4 | 43.6 | 29.4 | 35.7 | 26.9 | 29.4 | 62.4 | 39.0 | 49.6 | 64.1 | 40.4 |
| 16 | 24.5 | 44.9 | 30.5 | 37.5 | 26.9 | 28.8 | 63.8 | 39.3 | 52.1 | 66.6 | 41.5 |
| 20 | 25.3 | 46.5 | 30.2 | 38.2 | 27.8 | 31.0 | 66.9 | 39.9 | 51.9 | 66.4 | 42.4 |
| Raw + REFINEX | | | | | | | | | | | |
| 4 | 24.1 | 40.1 | 26.5 | 30.5 | 26.2 | 27.0 | 60.4 | 38.3 | 50.7 | 56.5 | 38.0 |
| 8 | 23.9 | 41.4 | 29.3 | 34.8 | 26.6 | 27.8 | 63.2 | 38.9 | 51.3 | 63.5 | 40.1 |
| 12 | 23.1 | 43.0 | 29.5 | 37.0 | 26.9 | 28.6 | 65.5 | 40.4 | 52.2 | 64.5 | 41.1 |
| 16 | 23.3 | 45.0 | 31.1 | 37.9 | 27.1 | 29.8 | 65.7 | 40.1 | 50.4 | 65.5 | 41.6 |
| 20 | 25.2 | 45.9 | 32.0 | 39.4 | 27.5 | 31.2 | 66.3 | 41.0 | 52.5 | 68.3 | 42.9 |
| Go | | | | | | | | | | | |
| 4 | 23.2 | 40.4 | 27.6 | 32.2 | 25.9 | 26.6 | 60.1 | 39.9 | 51.5 | 55.9 | 38.3 |
| 8 | 23.5 | 41.4 | 27.6 | 35.4 | 26.6 | 27.8 | 63.6 | 39.4 | 51.2 | 62.0 | 39.8 |
| 12 | 23.7 | 43.5 | 27.7 | 38.3 | 26.6 | 26.4 | 65.3 | 39.0 | 52.1 | 64.2 | 40.7 |
| 16 | 24.7 | 43.9 | 28.2 | 39.1 | 27.0 | 28.7 | 66.7 | 40.4 | 51.5 | 65.9 | 41.6 |
| 20 | 24.3 | 45.1 | 28.6 | 39.7 | 27.1 | 28.4 | 66.7 | 39.4 | 50.9 | 66.9 | 41.7 |
| Go + ProX | | | | | | | | | | | |
| 4 | 24.9 | 39.1 | 26.2 | 30.5 | 26.0 | 25.6 | 61.5 | 38.6 | 51.0 | 55.2 | 37.9 |
| 8 | 23.8 | 42.9 | 28.6 | 35.2 | 26.9 | 28.4 | 64.3 | 38.0 | 50.6 | 62.0 | 40.1 |
| 12 | 24.8 | 43.6 | 29.6 | 38.8 | 27.2 | 28.6 | 66.2 | 39.6 | 51.5 | 65.1 | 41.5 |
| 16 | 23.7 | 46.6 | 30.5 | 40.1 | 27.5 | 29.0 | 65.9 | 38.8 | 52.5 | 67.4 | 42.2 |
| 20 | 25.0 | 46.0 | 30.9 | 41.0 | 27.9 | 30.4 | 66.7 | 38.4 | 51.5 | 68.4 | 42.6 |
| Go + REFINEX | | | | | | | | | | | |
| 4 | 23.6 | 39.2 | 26.9 | 31.1 | 25.7 | 26.2 | 61.6 | 39.4 | 51.4 | 58.2 | 38.3 |
| 8 | 23.5 | 43.0 | 28.7 | 36.6 | 26.9 | 31.4 | 63.4 | 39.4 | 53.0 | 61.4 | 40.7 |
| 12 | 24.6 | 46.2 | 29.5 | 40.2 | 27.8 | 28.2 | 66.4 | 38.5 | 52.3 | 66.6 | 42.0 |
| 16 | 25.2 | 46.8 | 29.5 | 40.8 | 27.7 | 28.4 | 66.9 | 40.3 | 51.8 | 67.2 | 42.5 |
| 20 | 25.0 | 48.2 | 30.4 | 41.4 | 28.1 | 29.0 | 66.7 | 40.7 | 52.8 | 69.3 | 43.2 |
| C4 | | | | | | | | | | | |
| 4 | 21.2 | 37.1 | 27.1 | 30.1 | 25.8 | 28.6 | 59.5 | 38.7 | 51.2 | 56.8 | 37.7 |
| 8 | 23.1 | 40.4 | 28.2 | 35.5 | 27.2 | 28.0 | 62.8 | 37.4 | 50.5 | 62.5 | 40.5 |
| 12 | 24.7 | 44.8 | 30.4 | 37.2 | 27.0 | 28.2 | 65.0 | 39.1 | 51.3 | 67.1 | 40.8 |
| 16 | 24.1 | 44.2 | 29.1 | 39.1 | 27.5 | 28.8 | 66.2 | 38.7 | 51.4 | 66.9 | 41.9 |
| 20 | 25.3 | 44.2 | 30.9 | 39.3 | 27.4 | 29.4 | 66.5 | 39.5 | 51.3 | 67.3 | 42.1 |
| C4 + ProX | | | | | | | | | | | |
| 4 | 23.3 | 40.1 | 26.6 | 29.9 | 26.2 | 27.4 | 62.2 | 37.0 | 49.7 | 55.0 | 37.7 |
| 8 | 24.8 | 42.2 | 29.2 | 34.6 | 27.0 | 28.6 | 65.1 | 38.5 | 52.4 | 62.1 | 40.5 |
| 12 | 25.0 | 42.6 | 30.2 | 35.9 | 27.0 | 28.4 | 63.9 | 38.5 | 51.0 | 65.1 | 40.8 |
| 16 | 24.8 | 45.3 | 30.3 | 39.2 | 27.9 | 29.6 | 66.7 | 38.4 | 52.0 | 65.0 | 41.9 |
| 20 | 25.1 | 45.5 | 31.7 | 41.1 | 27.9 | 29.4 | 66.6 | 40.1 | 51.4 | 66.6 | 42.5 |
| C4 + REFINEX | | | | | | | | | | | |
| 4 | 21.8 | 39.3 | 27.2 | 30.7 | 25.7 | 25.2 | 62.4 | 39.1 | 50.5 | 59.3 | 38.1 |
| 8 | 23.9 | 41.6 | 30.0 | 34.2 | 26.7 | 28.2 | 62.8 | 39.4 | 50.2 | 60.5 | 39.7 |
| 12 | 23.9 | 43.3 | 31.7 | 37.9 | 26.8 | 28.6 | 65.9 | 39.2 | 50.3 | 67.2 | 41.5 |
| 16 | 24.3 | 44.7 | 31.4 | 40.1 | 27.7 | 29.4 | 67.3 | 38.0 | 50.4 | 67.5 | 42.1 |
| 20 | 24.8 | 44.5 | 31.6 | 41.2 | 28.0 | 30.4 | 68.0 | 39.9 | 50.5 | 68.2 | 42.7 |

Table 13: Full evaluation results (2/2) on a **750M** pretrained model across different downstream tasks, with varying numbers of training tokens used during pretraining.

| #token | ARC-C | ARC-E | CSQA | HellaSwag | MMLU | OBQA | PiQA | SIQA | WinoG | SciQ | AVG |
|--------|-------|-------|------|-----------|------|------|------|------|-------|------|-----|
| | | | | | Fw | | | | | | |
| 4 | 21.4 | 38.1 | 27.8 | 29.3 | 25.3 | 25.4 | 61.0 | 38.7 | 50.2 | 57.1 | 37.4 |
| 8 | 23.3 | 41.9 | 28.5 | 32.2 | 26.6 | 27.0 | 62.8 | 39.3 | 49.9 | 62.9 | 39.4 |
| 12 | 25.1 | 44.5 | 29.9 | 35.6 | 27.2 | 27.2 | 65.0 | 38.8 | 51.1 | 64.9 | 40.9 |
| 16 | 24.6 | 44.3 | 30.0 | 37.9 | 27.1 | 28.6 | 65.3 | 40.5 | 49.8 | 67.9 | 41.6 |
| 20 | 25.1 | 45.6 | 31.6 | 39.9 | 27.2 | 27.6 | 66.6 | 39.3 | 49.9 | 66.2 | 41.9 |
| | | | | | Fw + ProX | | | | | | |
| 4 | 23.7 | 37.5 | 26.2 | 30.9 | 26.2 | 27.8 | 61.3 | 38.9 | 49.4 | 57.7 | 38.0 |
| 8 | 24.9 | 42.0 | 28.9 | 33.8 | 26.6 | 26.0 | 65.4 | 39.9 | 49.7 | 62.6 | 40.0 |
| 12 | 26.8 | 42.7 | 29.9 | 37.1 | 26.8 | 29.2 | 65.7 | 39.8 | 49.8 | 65.0 | 41.3 |
| 16 | 26.4 | 44.5 | 30.0 | 38.3 | 27.5 | 29.0 | 65.9 | 39.3 | 51.8 | 66.8 | 42.0 |
| 20 | 25.5 | 44.7 | 30.8 | 39.0 | 27.9 | 29.2 | 67.4 | 39.5 | 51.3 | 67.5 | 42.3 |
| | | | | | Fw + REFINEX | | | | | | |
| 4 | 22.3 | 39.0 | 27.3 | 30.3 | 26.2 | 25.0 | 61.1 | 38.3 | 51.6 | 57.3 | 37.8 |
| 8 | 24.6 | 40.4 | 30.2 | 34.0 | 26.3 | 26.4 | 64.3 | 38.7 | 51.4 | 62.0 | 39.8 |
| 12 | 25.6 | 44.9 | 31.8 | 37.2 | 26.7 | 27.4 | 65.1 | 39.4 | 50.1 | 64.3 | 41.2 |
| 16 | 25.7 | 46.5 | 29.7 | 38.9 | 26.8 | 30.7 | 65.8 | 39.8 | 49.6 | 64.9 | 41.8 |
| 20 | 26.3 | 47.4 | 31.4 | 40.1 | 27.6 | 30.8 | 66.6 | 39.6 | 51.5 | 65.4 | 42.7 |
| | | | | | Comb | | | | | | |
| 4 | 22.3 | 40.9 | 27.0 | 30.7 | 26.3 | 28.2 | 62.0 | 38.6 | 50.7 | 57.6 | 38.4 |
| 8 | 23.6 | 41.8 | 28.5 | 34.4 | 25.7 | 28.2 | 63.8 | 38.3 | 50.4 | 62.9 | 39.8 |
| 12 | 23.4 | 43.2 | 30.2 | 37.5 | 26.6 | 29.0 | 65.9 | 39.3 | 52.4 | 65.2 | 41.3 |
| 16 | 23.4 | 44.2 | 31.4 | 40.4 | 27.1 | 27.8 | 66.4 | 39.5 | 52.8 | 66.5 | 41.9 |
| 20 | 24.3 | 44.7 | 31.0 | 40.4 | 27.1 | 28.6 | 66.2 | 39.0 | 53.3 | 66.5 | 42.1 |
| | | | | | Comb + ProX | | | | | | |
| 4 | 24.2 | 39.9 | 27.7 | 32.7 | 26.4 | 24.2 | 60.2 | 37.6 | 51.9 | 55.9 | 38.1 |
| 8 | 24.1 | 42.6 | 28.7 | 35.1 | 26.7 | 26.2 | 64.1 | 38.9 | 52.4 | 63.0 | 40.2 |
| 12 | 24.0 | 42.4 | 31.1 | 38.0 | 26.9 | 27.0 | 66.4 | 39.8 | 50.4 | 63.4 | 40.9 |
| 16 | 25.0 | 43.1 | 31.5 | 39.3 | 27.4 | 30.0 | 67.3 | 40.1 | 52.3 | 66.2 | 42.2 |
| 20 | 24.8 | 45.4 | 31.7 | 41.1 | 27.8 | 29.4 | 66.7 | 39.5 | 51.7 | 65.8 | 42.4 |
| | | | | | Comb + REFINEX | | | | | | |
| 4 | 23.1 | 39.1 | 27.5 | 30.6 | 26.1 | 24.2 | 62.1 | 39.3 | 50.0 | 57.8 | 38.0 |
| 8 | 25.3 | 42.5 | 28.6 | 35.5 | 26.8 | 27.8 | 65.1 | 39.3 | 52.2 | 62.4 | 40.6 |
| 12 | 25.4 | 44.6 | 29.8 | 38.1 | 26.9 | 29.0 | 66.1 | 39.8 | 52.8 | 65.0 | 41.7 |
| 16 | 25.5 | 45.9 | 30.6 | 40.1 | 27.6 | 29.0 | 66.2 | 38.1 | 54.5 | 66.0 | 42.3 |
| 20 | 25.5 | 46.6 | 31.4 | 42.1 | 27.5 | 28.0 | 68.0 | 39.6 | 53.2 | 65.9 | 42.8 |
| | | | | | ProX-D | | | | | | |
| 4 | 21.4 | 38.1 | 27.8 | 29.3 | 25.3 | 25.4 | 61.0 | 38.7 | 50.2 | 57.1 | 37.4 |
| 8 | 23.3 | 41.9 | 28.5 | 32.2 | 26.6 | 27.0 | 62.8 | 39.3 | 49.9 | 62.9 | 39.4 |
| 12 | 25.1 | 44.5 | 29.9 | 35.6 | 27.2 | 27.2 | 65.0 | 38.8 | 51.1 | 64.9 | 40.9 |
| 16 | 24.6 | 44.3 | 30.0 | 37.9 | 27.1 | 28.6 | 65.3 | 40.5 | 49.8 | 67.9 | 41.6 |
| 20 | 25.1 | 45.6 | 31.6 | 39.9 | 27.2 | 27.6 | 66.6 | 39.3 | 49.9 | 67.3 | 42.0 |
| | | | | | ProX-D + ProX | | | | | | |
| 4 | 23.3 | 44.4 | 27.5 | 32.2 | 27.1 | 28.8 | 61.4 | 37.8 | 49.9 | 61.7 | 39.4 |
| 8 | 26.1 | 46.7 | 27.3 | 35.3 | 28.0 | 27.2 | 63.3 | 37.4 | 50.7 | 64.8 | 40.7 |
| 12 | 26.1 | 50.9 | 28.8 | 38.2 | 28.2 | 30.6 | 64.1 | 38.3 | 49.9 | 69.0 | 42.4 |
| 16 | 27.1 | 51.7 | 30.2 | 40.7 | 28.5 | 33.8 | 66.0 | 38.6 | 50.1 | 69.3 | 43.6 |
| 20 | 27.2 | 51.0 | 30.1 | 41.2 | 28.9 | 30.4 | 65.9 | 39.4 | 50.8 | 70.2 | 43.5 |
| | | | | | ProX-D + REFINEX | | | | | | |
| 4 | 24.3 | 44.2 | 26.1 | 31.5 | 26.8 | 25.6 | 60.4 | 38.4 | 51.6 | 61.4 | 39.0 |
| 8 | 26.5 | 47.9 | 29.0 | 36.3 | 27.7 | 29.2 | 64.0 | 38.8 | 50.4 | 64.4 | 41.4 |
| 12 | 26.0 | 50.3 | 29.4 | 38.0 | 28.7 | 30.4 | 64.2 | 39.3 | 51.0 | 66.8 | 42.4 |
| 16 | 27.7 | 51.7 | 29.8 | 40.9 | 28.9 | 31.2 | 66.2 | 39.7 | 52.8 | 68.8 | 43.8 |
| 20 | 28.7 | 53.2 | 30.8 | 41.7 | 29.6 | 31.8 | 67.8 | 39.9 | 51.6 | 70.9 | 44.7 |

Table 14: Cases study illustrating the deletion-only constraint in REFINEX. E2E modifies stylistic or lexical content, while REFINEX performs only removals, preserving imperfections.

| Case 1: Deletion-Only Constraint |
|---|

**Raw Text:**
Climate change has become one of the most urgent issues facing our planet. Many scientists agree that human activities are significantly affecting the **enviroment** in a negative way.
Click here to BUY NOW: www.greendeals.fake/discount
In recent years, more people have taken interest in renewable energy sources such as solar and wind power.
$$$$###!!! <broken tag> // sdlfkj243==
This transition could greatly reduce our dependence on fossil fuels.
Visit ENVYBOOST.SITE to earn $$$ from home! ##

**Refined by E2E:**
Climate change has become one of the most urgent issues facing our planet. Many scientists agree that human activities are significantly affecting the **environment** in a negative way.
In recent years, more people have taken interest in renewable energy sources such as solar and wind power.
This transition could greatly reduce our dependence on fossil fuels.

**Refined by REFINEX:**
Climate change has become one of the most urgent issues facing our planet. Many scientists agree that human activities are significantly affecting the **enviroment** in a negative way.
In recent years, more people have taken interest in renewable energy sources such as solar and wind power.
This transition could greatly reduce our dependence on fossil fuels.

| Case 2: Deletion vs. Stylistic Rewriting |
|---|

**Raw Text:**
The study results show that people **don't sleep well** when exposed to high levels of blue light at night. Buy blue-light blockers now! Limited-time offer: www.sleepmagic.io
This can lead to issues with memory and focus. sldkfj234##!! <meta-junk> watch till the end It's better to use dim lights or avoid screens before going to bed.

**Refined by E2E:**
The study results indicate that people **experience impaired sleep quality** when exposed to high levels of blue light at night.
This can lead to issues with memory and focus. It's better to use dim lights or avoid screens before going to bed.

**Refined by REFINEX:**
The study results show that people **don't sleep well** when exposed to high levels of blue light at night.
This can lead to issues with memory and focus. It's better to use dim lights or avoid screens before going to bed.

| Case 3: Unintended Stylistic Change |
|---|

**Raw Text:**
Many students **didn't really understand** the topic after the lecture. Watch our sponsors at www.ads-university.biz
This suggests that the teaching method might need improvement. asd8127&# noise fragment Learn Crypto in 2 weeks!

**Refined by E2E:**
Many students **struggled to grasp** the topic after the lecture. This suggests that the teaching method might need improvement.

**Refined by REFINEX:**
Many students **didn't really understand** the topic after the lecture. This suggests that the teaching method might need improvement.

Table 15: Cases (1/5) after applying ProX and REFINEX. Text in red indicates low-value content to be removed. "..." denotes omitted content due to limited space.

| Case 1 |
| --- |

**Raw Text:**

...
Lerner's engaging prose and robust idealism cast laboratory methods in a positive light, yet he does not overlook the dark side of "experimentation towards what works" (12). At the end of chapter eight, he acknowledges that experimentation entails failure. "What I am urging here is for reformers ... to have the courage to fail—but then learn from those failures and mount a new experiment rather than revert to the status quo" (187). Lerner makes this bold call, though he is well aware that the structure of higher education is not conducive to it. Failure is an anathema in higher education, particularly in tight budgetary times. After observing that literacy activities in the early grades often illustrate laboratory methods of learning, he remarks, "The challenge, certainly, is how to retain the play that marks these early efforts at language learning in a system that often does not believe in playing around" (196). The call to pursue laboratory methods may be less appealing when it is understood as a call to embrace failure, even if only in the short-term.
Return to Composition Forum 21 table of contents.  desktop is booked in for repair it goes into a queue to await the services of a technician.
($cookies, $Dallas)  DBrandeis-Bardin campusresh, reflect and enjoy each other's company!
We offer multiple venues for your Family Reunion, Wedding Reception or Rehearsal Dinner. Contact us.
...

**Refined by Prox:**

...
Lerner's engaging prose and robust idealism cast laboratory methods in a positive light, yet he does not overlook the dark side of "experimentation towards what works" (12). At the end of chapter eight, he acknowledges that experimentation entails failure. "What I am urging here is for reformers ... to have the courage to fail—but then learn from those failures and mount a new experiment rather than revert to the status quo" (187). Lerner makes this bold call, though he is well aware that the structure of higher education is not conducive to it. Failure is an anathema in higher education, particularly in tight budgetary times. After observing that literacy activities in the early grades often illustrate laboratory methods of learning, he remarks, "The challenge, certainly, is how to retain the play that marks these early efforts at language learning in a system that often does not believe in playing around" (196). The call to pursue laboratory methods may be less appealing when it is understood as a call to embrace failure, even if only in the short-term.
Return to Composition Forum 21 table of contents.  desktop is booked in for repair it goes into a queue to await the services of a technician.
($cookies, $Dallas)  DBrandeis-Bardin campusresh, reflect and enjoy each other's company!
...

**Refined by REFINEX:**

...
Lerner's engaging prose and robust idealism cast laboratory methods in a positive light, yet he does not overlook the dark side of "experimentation towards what works" (12). At the end of chapter eight, he acknowledges that experimentation entails failure. "What I am urging here is for reformers ... to have the courage to fail—but then learn from those failures and mount a new experiment rather than revert to the status quo" (187). Lerner makes this bold call, though he is well aware that the structure of higher education is not conducive to it. Failure is an anathema in higher education, particularly in tight budgetary times. After observing that literacy activities in the early grades often illustrate laboratory methods of learning, he remarks, "The challenge, certainly, is how to retain the play that marks these early efforts at language learning in a system that often does not believe in playing around" (196). The call to pursue laboratory methods may be less appealing when it is understood as a call to embrace failure, even if only in the short-term.
...

Table 16: Cases (2/5) after applying ProX and REFINEX. Text in red indicates low-value content to be removed. ". . ." denotes omitted content due to limited space.

| Case 2 |
|---|

**Raw Text:**

. . .

June 24, 2016
STORE | CHECKOUT/CART
EXCLUSIVE // Listen to Stage Kids' new album 'Intra Mental' in full
Five years after their full-length release, Killer Tofu, the San Diego-based instrumental band Stage Kids have reemerged with their latest masterpiece, Intra Mental. Two new members (keyboard and electronics) have joined the quintet since 2011, resulting in more complex and dynamic soundscapes. But the majority of us are summing this one up in one word: definitive. Stage Kids waste no time introducing their new sound with the opening track "Delaylay". The atmospheric synth and guitar give way to a barrage of start/stop rhythms that jump from idea to idea, frequently straying from the tonic of the song, but never too far from comfort. In "Face First" the rapid staccato of the guitar and stabbing piano keys create an almost glitchy feel that evolves into frenetically tapped phrases typical of bands like Rooftops, Invalids, and Floral. In "Pulsewave", "The Noise After", and "Connections", Stage Kids re-direct their song writing towards electronic sensibilities.
Intra Mental comes out this weekend. You can get all the info on their Bandcamp page. You can also keep in touch with Stage Kids on their Facebook page.
Related exclusive fecking bahamas full intra mental stage kids stream Share On Tweet
Previous ArticleEXCLUSIVE // Dumb Waiter premier new track 'Cheesevader'
Stephen Kemp Contributor Related Posts
. . .

**Refined by Prox:**

. . .

June 24, 2016
Five years after their full-length release, Killer Tofu, the San Diego-based instrumental band Stage Kids have reemerged with their latest masterpiece, Intra Mental. Two new members (keyboard and electronics) have joined the quintet since 2011, resulting in more complex and dynamic soundscapes. But the majority of us are summing this one up in one word: definitive. Stage Kids waste no time introducing their new sound with the opening track "Delaylay". The atmospheric synth and guitar give way to a barrage of start/stop rhythms that jump from idea to idea, frequently straying from the tonic of the song, but never too far from comfort. In "Face First" the rapid staccato of the guitar and stabbing piano keys create an almost glitchy feel that evolves into frenetically tapped phrases typical of bands like Rooftops, Invalids, and Floral. In "Pulsewave", "The Noise After", and "Connections", Stage Kids re-direct their song writing towards electronic sensibilities.
Intra Mental comes out this weekend. You can get all the info on their Bandcamp page. You can also keep in touch with Stage Kids on their Facebook page.
. . .

**Refined by REFINEX:**

. . .

Five years after their full-length release, Killer Tofu, the San Diego-based instrumental band Stage Kids have reemerged with their latest masterpiece, Intra Mental. Two new members (keyboard and electronics) have joined the quintet since 2011, resulting in more complex and dynamic soundscapes. But the majority of us are summing this one up in one word: definitive. Stage Kids waste no time introducing their new sound with the opening track "Delaylay". The atmospheric synth and guitar give way to a barrage of start/stop rhythms that jump from idea to idea, frequently straying from the tonic of the song, but never too far from comfort. In "Face First" the rapid staccato of the guitar and stabbing piano keys create an almost glitchy feel that evolves into frenetically tapped phrases typical of bands like Rooftops, Invalids, and Floral. In "Pulsewave", "The Noise After", and "Connections", Stage Kids re-direct their song writing towards electronic sensibilities.
. . .

Table 17: Cases (3/5) after applying ProX and REFINEX. Text in red indicates low-value content to be removed. "..." denotes omitted content due to limited space.

| Case 3 |
| --- |

**Raw Text:**
Atlanta Palm Beach Tampa / St. Pete
My Name is Maryan – Docent-Led Exhibition Tour
TAGS: Art Museum Tour
Sun 08/14/2022
Register for a tour led by MOCA docent Dr. Helen Sachs Chaset. Dr. Chaset is an educator with more than 45 years of experience in administration, professional development, and program development. She is also a board member of Miami-Dade Holocaust Survivors, the Jewish Community Services of South Florida and the Center for the Advancement of Jewish Education. Dr. Chaset is the daughter of two Holocaust Survivors and was born in a displaced persons camp in Hannover, Germany.
Time 11:30 a.m.
Venue Museum of Contemporary Art North Miami
Address 770 N.E. 125 Street
North Miami, FL 33161 GET DIRECTIONS
The Palm Miami
BOURBON STEAK
Nature Photography at Deering Estate - March
Art on the Plaza: "Clint and April"
Bird Walk at Deering Estate
OUTshine LGBTQ+ Film Festival | Fort Lauderdale Edition

**Refined by Prox:**
My Name is Maryan – Docent-Led Exhibition Tour
TAGS: Art Museum Tour
Sun 08/14/2022
Register for a tour led by MOCA docent Dr. Helen Sachs Chaset. Dr. Chaset is an educator with more than 45 years of experience in administration, professional development, and program development. She is also a board member of Miami-Dade Holocaust Survivors, the Jewish Community Services of South Florida and the Center for the Advancement of Jewish Education. Dr. Chaset is the daughter of two Holocaust Survivors and was born in a displaced persons camp in Hannover, Germany.
Time 11:30 a.m.
Venue Museum of Contemporary Art North Miami
Address 770 N.E. 125 Street
North Miami, FL 33161 GET DIRECTIONS
The Palm Miami
BOURBON STEAK
Nature Photography at Deering Estate - March
Art on the Plaza: "Clint and April"
Bird Walk at Deering Estate

**Refined by REFINEX:**
My Name is Maryan – Docent-Led Exhibition Tour
TAGS: Art Museum Tour
Sun 08/14/2022
Register for a tour led by MOCA docent Dr. Helen Sachs Chaset. Dr. Chaset is an educator with more than 45 years of experience in administration, professional development, and program development. She is also a board member of Miami-Dade Holocaust Survivors, the Jewish Community Services of South Florida and the Center for the Advancement of Jewish Education. Dr. Chaset is the daughter of two Holocaust Survivors and was born in a displaced persons camp in Hannover, Germany.
Time 11:30 a.m.
Venue Museum of Contemporary Art North Miami
Address 770 N.E. 125 Street
North Miami, FL 33161

Table 18: Cases (4/5) after applying ProX and REFINEX. Text in red indicates low-value content to be removed. "..." denotes omitted content due to limited space.

| Case 4 |
|---|

**Raw Text:**

Justin Iveland and Claude Weisbuch. Direct measurement of Auger electrons emitted from a semiconductor light-emitting diode under electrical injection: identification of the dominant mechanism for efficiency droop. *Phys. Rev. Lett.*, 110, 177406, 2013.

Fred Jendrzejewski, Alain Bernard, Killian Muller, Patrick Cheinet, Vincent Josse, Marie Piraud, Luca Pezzé, Laurent Sanchez-Palencia, Alain Aspect, and Philippe Bouyer. Threeture Physics 8, 398, 2012.

Fred Jendrzejewski, Killian Muller, Jérémie Richard, Aditya Date, Thomas Plisson, Philippe Bouyer, Alain Aspect, and Vincent Josse. Coherent Backscattering of Ultracold Atoms. *Physical Review Letters* 109 (19), 2012.

Juliette Billy, Vincent Josse, Zhanchun C. Zuo, Alain Bernard, Ben Hambrecht, Pierre Lugan, David Clément, Laurent Sanchez-Palencia, Philippe Bouyer, and Alaine 453, 891, 2008.

S. Félix, M. Asch, M. Filoche, and B. Sapoval. Localization and increased damping in irregular acoustic cavities. *Journal of Science and Vibration* 299, 965-976, 2007.

Luis A. CaffarYves Meyer and Ronald R. Coifman. Wavelets: Calderón–Zygmund and Multilinear Operators. Number 48 in7.fi.

 Another woman who worked at the ABC's Toowong office in Brisbane has been diagnosed with breast cancer. LEIGH SALES, PRESENTER: Another woman who worked at the ABCTests carried out on the site failed to uncover a cause of the cluster.  How do I insert the citation? According to Smith (2016) sup.14. Sustainability can be defined as a socio-ecological process (Smith, 2016). Cite both names everytime you refer to their work. You must cite all the authors' names the first time you refer to that work. The work you submit for grading must be your own. If you use work from other sources, you must acknowledge it properly. This also applies to online sources such as the web. Your work will be assession tools. Visit the Academic integrity and copyright website for further information..  Whats the change the fans on laptop (approximately)? 2m experience with how was plugged into an outlet. It isn't causing any with to recover my files with chips called? After a few minutes update (64) + about 5 3020 brands than Intel?

**Refined by Prox:**

Justin Iveland and Claude Weisbuch. Direct measurement of Auger electrons emitted from a semiconductor light-emitting diode under electrical injection: identification of the dominant mechanism for efficiency droop. *Phys. Rev. Lett.*, 110, 177406, 2013.

Fred Jendrzejewski, Alain Bernard, Killian Muller, Patrick Cheinet, Vincent Josse, Marie Piraud, Luca Pezzé, Laurent Sanchez-Palencia, Alain Aspect, and Philippe Bouyer. Threeture Physics 8, 398, 2012.

Fred Jendrzejewski, Killian Muller, Jérémie Richard, Aditya Date, Thomas Plisson, Philippe Bouyer, Alain Aspect, and Vincent Josse. Coherent Backscattering of Ultracold Atoms. *Physical Review Letters* 109 (19), 2012.

Juliette Billy, Vincent Josse, Zhanchun C. Zuo, Alain Bernard, Ben Hambrecht, Pierre Lugan, David Clément, Laurent Sanchez-Palencia, Philippe Bouyer, and Alaine 453, 891, 2008.

S. Félix, M. Asch, M. Filoche, and B. Sapoval. Localization and increased damping in irregular acoustic cavities. *Journal of Science and Vibration* 299, 965-976, 2007.

Luis A. CaffarYves Meyer and Ronald R. Coifman. Wavelets: Calderón–Zygmund and Multilinear Operators. Number 48 in7.fi.

 Another woman who worked at the ABC's Toowong office in Brisbane has been diagnosed with breast cancer. LEIGH SALES, PRESENTER: Another woman who worked at the ABCTests carried out on the site failed to uncover a cause of the cluster.  How do I insert the citation? According to Smith (2016) sup.14. Sustainability can be defined as a socio-ecological process (Smith, 2016). Cite both names everytime you refer to their work.

**Refined by REFINEX:**

Justin Iveland and Claude Weisbuch. Direct measurement of Auger electrons emitted from a semiconductor light-emitting diode under electrical injection: identification of the dominant mechanism for efficiency droop. *Phys. Rev. Lett.*, 110, 177406, 2013.

Fred Jendrzejewski, Alain Bernard, Killian Muller, Patrick Cheinet, Vincent Josse, Marie Piraud, Luca Pezzé, Laurent Sanchez-Palencia, Alain Aspect, and Philippe Bouyer. Threeture Physics 8, 398, 2012.

Fred Jendrzejewski, Killian Muller, Jérémie Richard, Aditya Date, Thomas Plisson, Philippe Bouyer, Alain Aspect, and Vincent Josse. Coherent Backscattering of Ultracold Atoms. *Physical Review Letters* 109 (19), 2012.

Juliette Billy, Vincent Josse, Zhanchun C. Zuo, Alain Bernard, Ben Hambrecht, Pierre Lugan, David Clément, Laurent Sanchez-Palencia, Philippe Bouyer, and Alaine 453, 891, 2008.

S. Félix, M. Asch, M. Filoche, and B. Sapoval. Localization and increased damping in irregular acoustic cavities. *Journal of Science and Vibration* 299, 965-976, 2007.

Table 19: Cases (5/5) after applying ProX and REFINEX. Text in red indicates low-value content to be removed. "..." denotes omitted content due to limited space.

| Case 5 |
| --- |

**Raw Text:**

...
Comments are turned off. Learn more
Description
Subscribe for more movie scenes! Shrek 2 - Livin' la Vida Loca scene
Show less Show more
Watch on YouTube
Animation • 2004 • 1 hr 32 min
English audio
BUY OR RENT
Happily ever after never seemed so far far away when a trip to meet the in-laws turns into a hilariously twisted adventure for Shrek (Mike Myers) and Fiona (Cameron Diaz). With the help of his faithful Donkey (Eddie Murphy), Shrek takes on a potion-brewing Fairy Godmother, the pompous Prince Charming (Rupert Everett), and the ogre-killer, Puss In Boots (Antonio Banderas) who's a pussycat at heart.
Save the Enemy | Spookiz | Cartoons for Kids | WildBrain Kids
WildBrain Kids
...

**Refined by Prox:**

...
Comments are turned off. Learn more
Description
Subscribe for more movie scenes! Shrek 2 - Livin' la Vida Loca scene
Show less Show more
Watch on YouTube
Animation • 2004 • 1 hr 32 min
English audio
BUY OR RENT
Happily ever after never seemed so far far away when a trip to meet the in-laws turns into a hilariously twisted adventure for Shrek (Mike Myers) and Fiona (Cameron Diaz). With the help of his faithful Donkey (Eddie Murphy), Shrek takes on a potion-brewing Fairy Godmother, the pompous Prince Charming (Rupert Everett), and the ogre-killer, Puss In Boots (Antonio Banderas) who's a pussycat at heart.
Save the Enemy | Spookiz | Cartoons for Kids | WildBrain Kids
WildBrain Kids
...

**Refined by REFINEX:**

...
Happily ever after never seemed so far far away when a trip to meet the in-laws turns into a hilariously twisted adventure for Shrek (Mike Myers) and Fiona (Cameron Diaz). With the help of his faithful Donkey (Eddie Murphy), Shrek takes on a potion-brewing Fairy Godmother, the pompous Prince Charming (Rupert Everett), and the ogre-killer, Puss In Boots (Antonio Banderas) who's a pussycat at heart.
...

