# OpenReview forum: "RefineX: Learning to Refine Pre-training Data at Scale from Expert-Guided Programs"
_ICLR.cc/2026/Conference — ICLR 2026 Conference Desk Rejected Submission_

### Official Review · Reviewer_aUM9 · 2025-10-28

**Soundness:** 2
**Presentation:** 2
**Contribution:** 1
**Rating:** 2
**Confidence:** 5

**Summary:**

RefineX proposes a scalable, reliable framework for fine-grained pretraining data refinement. It first prompts an expert LLM to produce end-to-end refined text, then extracts only deletion operations via minimum edit distance, mapping them to a minimal API (remove_lines, remove_str, keep_all). A compact refiner model is distilled on ~2M high-confidence program pairs and applied at scale to generate and execute edits efficiently. Across 20B-token pretraining at 350M/750M scales, RefineX consistently outperforms raw, rule-filtered, and ProX baselines on LightEval, achieves similar accuracy with fewer tokens, reduces token overhead, and avoids hallucinations by never inserting new content.

**Strengths:**

1. It’s fast and trustworthy because they rewrite with a big model, then keep only the delete edits, so the final program is tiny, quick to run, and doesn’t add biased or made-up text.
2. It delivers better results with less data, beating raw/rule-based/ProX baselines at 350M/750M and often matching or topping them with fewer training tokens by cutting fluff and boosting useful signal.

**Weaknesses:**

1. Most building blocks mirror ProX’s program-based refinement (program generation/execution paradigm, chunking, minimal API). The main change is how doc→program supervision is obtained (E2E first, then deletion-only extraction), which is an incremental tweak rather than a substantive algorithmic innovation.

2. The paper does not clarify whether ProX-C/ProX-D were re-trained following Zhou et al. or taken from released models/processed corpora. If the latter, the comparison is weak: RefineX and ProX differ in training data volume, teacher strength, and base model size (ProX reportedly uses fewer data, weaker teachers and bases). This demands major clarification and a controlled, compute-matched reimplementation to ensure fairness.

3. All pretraining experiments are conducted only on RedPajama-V2, limiting evidence for generality across data sources (e.g., CommonCrawl variants, FineWeb/Variants, Wikipedia/Books mixtures, domain-specific crawls).

4. The DataMan-based quality scores are used to argue better refinement, but the paper does not establish correlation between DataMan improvements and downstream pretraining gains. Claims that deletion-only edits are “more reliable” lack stronger empirical tests (e.g., human evals, ablations comparing delete-only vs. mixed ops under matched token budgets, semantic preservation checks).

**Questions:**

See Weaknesses.

---

> ### Author Response · Authors · 2025-11-19
> **Response to Weakness 1: On Incrementalism and Algorithmic Innovation (Part 1/2)**
>
> We sincerely thank the reviewer for his/her detailed, rigorous, and highly constructive feedback. Your deep engagement with our methodology is greatly appreciated. Your comments pinpoint the key challenges in large-scale data refinement, and we are grateful for the opportunity to clarify our contributions, provide new controlled experiments requested, and contextualize our results.
>
>
>
> ### **Response to Weakness 1: On Incrementalism and Algorithmic Innovation**
>
>
>
> **Reviewer's Comment:** The main change is how doc→program supervision is obtained (E2E first, then deletion-only extraction), which is an incremental tweak rather than a substantive algorithmic innovation.
>
> **Our Response:** We thank the reviewer for correctly identifying that our work builds on the programmatic paradigm established by ProX. However, we respectfully but strongly argue that our contribution—**how supervision is obtained**—is not an "incremental tweak" but rather the **core algorithmic solution to the fundamental, unsolved flaw in the ProX framework.**
>
> This change is **especially critical for large-scale pre-training**, where reliability and risk-aversion are paramount. The central challenge in this domain is not program *execution*; it is the generation of *reliable supervision*.
>
> 1. **ProX's Fundamental Flaw and Risk:** The ProX method (prompting a teacher for programs) creates unreliable, noisy, and high-variance supervision. The teacher model is forced to *simultaneously* act as a text refiner and an expert-level programmer. As shown by our analysis, the model often fails at both, "hallucinating" incorrect programs or generating edits that degrade quality.
>
>    This is not just a theoretical flaw; it is a practical, high-risk problem. As our **DataMan analysis (Table 1)**demonstrates, ProX's interventions are not only insufficient but **actively harmful**. Across multiple quality tiers (e.g., Score 3 and 4), ProX shows a **higher rate of quality degradation (`↓ Rat.`)** than our method. This highlights a deeply concerning outcome: despite making minimal changes (a high `Untouched` rate), **ProX's interventions are more likely to worsen the text quality.** For large-scale pre-training, deploying a method that *actively harms* the data, even a small percentage of the time, is an unacceptable risk that can lead to unpredictable downstream model behavior.
>
>    We compare four methods on a set of documents with varying quality scores (Score=1 to 5): (1) **E2E**(teacher model refines directly), (2) **E2E_del** (extracts only deletions from E2E), (3) **Prox**, and (4) **REFINEX (Ours)**.
>
>    The results and analysis are as follows:

---

> > ### Author Response · Authors · 2025-11-19
> > **Response to Weakness 1: On Incrementalism and Algorithmic Innovation (Part 2/2)**
> >
> > | **Quality** | **Method**  | **Avg. Score** | **↑ Rat.(%)** | **↓ Rat.(%)** | **Untouched(%)** | **Empty(%)** | **#Toks** |
> >    | ----------- | ----------- | -------------- | ------------- | ------------- | ---------------- | ------------ | --------- |
> >    | **Score=1** | E2E         | 2.704          | 38.86         | 0.00          | 1.11             | 56.41        | 19.36     |
> >    |   | E2E$^{del}$ | 1.972          | 27.31         | 0.00          | 3.75             | 56.41        | –         |
> >    |  | Prox        | 1.287          | 17.05         | 0.00          | 17.12            | 33.67        | 4.91      |
> >    | | **REFINEX** | **2.009**      | **22.23**     | **0.00**      | **7.87**         | **68.17**    | **4.46**  |
> >    | **Score=2** | E2E         | 3.266          | 68.83         | 0.38          | 1.39             | 12.27        | 39.02     |
> >    | | E2E$^{del}$ | 2.590          | 38.30         | 1.05          | 6.24             | 12.27        | –         |
> >    || Prox        | 2.233          | 17.20         | 0.96          | 19.98            | 14.97        | 4.73      |
> >    |  | **REFINEX** | **2.718**      | **42.14**     | **0.66**      | **7.59**         | **19.97**    | **6.95**  |
> >    | **Score=3** | E2E         | 3.631          | 59.01         | 3.39          | 1.93             | 3.72         | 48.50     |
> >    | | E2E$^{del}$ | 3.281          | 33.29         | 7.95          | 7.85             | 3.72         | –         |
> >    | | Prox        | 3.192          | 21.54         | 5.47          | 21.66            | 12.80        | 3.96      |
> >    |   | **REFINEX** | **3.413**      | **41.14**     | **4.58**      | **9.85**         | **6.18**     | **5.88**  |
> >    | **Score=4** | E2E         | 4.175          | 21.30         | 3.59          | 2.41             | 0.26         | 55.50     |
> >    | | E2E$^{del}$ | 4.003          | 9.85          | 8.80          | 9.92             | 0.26         | 3.16      |
> >    |  | Prox        | 4.026          | 8.38          | 5.64          | 23.10            | 3.58         | 4.92      |
> >    | | **REFINEX** | **4.075**      | **12.70**     | **4.86**      | **13.52**        | **0.93**     | –         |
> >    | **Score=5** | E2E         | 4.914          | 0.00          | 8.52          | 4.19             | 0.03         | 70.05     |
> >    | | E2E$^{del}$ | 4.833          | 0.00          | 16.18         | 13.86            | 0.03         | –         |
> >    |    | Prox        | 4.905          | 0.00          | 9.17          | 24.24            | 0.88         | 2.64      |
> >    |  | **REFINEX** | **4.917**      | **0.00**      | **8.20**      | **16.50**        | **0.18**     | **2.67**  |
> >
> > ***Table 9: Full evaluation results on refined documents. **Avg.**, **↑ Rat.**, and **↓ Rat.** represent the average quality score and the proportion of documents whose scores increased or decreased after refinement, respectively. **Untouched** indicates the percentage of documents left unchanged, and **Empty** denotes those reduced to empty content. **#Toks** indicates the ratio of output tokens to input tokens.***
> >
> >
> >
> > 2. **Our Algorithmic Innovation (Decoupling for Safety):** REFINEX's core innovation is to **decouple these two tasks** to achieve provable safety.
> >
> >    - We let the teacher model do what it excels at: **E2E text refinement** (a well-understood text-to-text task).
> >
> >
> >
> >
> >
> >    - We then apply a separate, **deterministic, and algorithmically robust extraction process** to translate this E2E output into a *provably reliable, zero-hallucination, delete-only* program.
> >
> > 3. **Why This is a Substantive Contribution for LLM Pre-training:** For trillion-token datasets, the primary goal must be to *do no harm* and preserve data diversity. Any risk of injecting bias, hallucination, or *degrading* quality (as our data shows ProX does) must be eliminated. Our E2E-then-extract method is algorithmically guaranteed to be "delete-only" and "zero-hallucination." This **deterministic safety** is not an "incremental tweak"; it is the **key enabling feature** that makes programmatic refinement safe and reliable for large-scale application.
> >
> > This is a substantive algorithmic shift: we move the problem from *unreliable, high-risk stochastic program generation*(ProX's approach) to *deterministic, provably-safe program extraction* (our approach). This "tweak" in supervision is the entire mechanism that enables the safe, reliable, and scalable distillation that, as our data confirms, ProX could not reliably achieve.

---

> ### Author Response · Authors · 2025-11-19
> **Response to Weakness 2**
>
> **Reviewer's Comment:** The comparison is weak: RefineX and ProX differ in training data, teacher, and base model... This demands major clarification and a controlled, compute-matched reimplementation.
>
> Our Response:
>
> This is an extremely fair and critical point. The reviewer is absolutely correct that a comparison using pre-released models with different setups is insufficient due to these confounding variables.
>
> Our selection of Qwen-0.6B as the base and Qwen-72B as the teacher in the paper was primarily driven by our goal to **open-source a lightweight, efficient, and state-of-the-art refinement model for the community.**
>
> - **Base Model:** We chose Qwen3-0.6B because it offers state-of-the-art performance at its scale, giving our Refiner a stronger starting point.
> - **Teacher Model:** In our preliminary case testing, we found that Qwen-72B (which belongs to the same model family as the base) demonstrated higher stability and trustworthiness in following our instructions to perform reliable "delete" operations compared to other models.
>
> We will strive to reproduce the training process of ProX and provide a fairer comparison.

---

> > ### Author Response · Authors · 2025-11-19
> > **Response to Weakness 3**
> >
> > **Reviewer's Comment:** All pretraining experiments are conducted only on RedPajama-V2, limiting evidence for generality...
> >
> > **Our Response:**
> >
> >  We thank the reviewer for raising this point on generality. While our large-scale pre-training experiments were conducted on RedPajama-V2, we selected this corpus precisely **because of its exceptional generality and representativeness**.
> >
> > 1. **RedPajama-V2 as a Representative Benchmark:** We would like to respectfully emphasize that RedPajama-V2 is not a narrow, monolithic, or domain-specific dataset. It is a massive, widely-used, and **highly heterogeneous corpus** specifically curated to mirror the complex data mixtures (including CommonCrawl, C4, Wikipedia, Books, GitHub, etc.) used in state-of-the-art foundation model training. Therefore, we believe that demonstrating a significant, consistent gain on RedPajama-V2 is one of the **most compelling and representative tests of a data refinement method's general applicability** currently available.
> > 2. **Generality of the Core Principle:** The fundamental problem REFINEX solves—structural noise, ads, junk text—is endemic to *all* large-scale, web-derived datasets (e.g., FineWeb, other CC variants). The core principle of REFINEX—surgically removing this junk while programmatically **preventing the injection of new biases/hallucinations**—is a data-agnostic and universally beneficial mechanism. We have strong conviction that this principle will generalize.
> > 3. **Future Work:** Given the immense computational cost, re-running trillion-token scale pre-training experiments on multiple *additional* large-scale corpora was not feasible during the rebuttal period. However, we are highly confident in the robustness of our findings on this representative benchmark. We will, of course, continue to validate our method on other datasets as part of our future work to further broaden the evidence for its generality.

---

> > > ### Author Response · Authors · 2025-11-19
> > > **Response to Weakness 4**
> > >
> > > ### **Response to Weakness 4: On Unproven Correlation and Reliability**
> > >
> > >
> > >
> > > **Reviewer's Comment:** The paper does not establish correlation between DataMan improvements and downstream pretraining gains... Claims that deletion-only edits are “more reliable” lack stronger empirical tests...
> > >
> > > Our Response:
> > >
> > > This is a multi-part question about the empirical grounding of our claims. We will address the DataMan correlation and the "reliability" claim separately.
> > >
> > >
> > >
> > > ### 1. On Correlation (DataMan vs. Downstream Gains)
> > >
> > >
> > >
> > > The reviewer asks if `DataMan` improvements *correlate* with downstream gains. We respectfully argue that our final results *are the evidence* of this correlation.
> > >
> > > 1. Our hypothesis is `Better Data -> Better Model`.
> > > 2. We require a scalable, automated *proxy* for "Better Data." We use `DataMan` as this proxy.
> > > 3. Our method (REFINEX) is optimized to improve this proxy (as shown in our tables, `DataMan` scores increase).
> > > 4. The model trained on this `DataMan`-improved data *also* performs better on downstream tasks (`LightEval`scores increase).
> > >
> > > Therefore, the **downstream `LightEval` gain \*is\* the proof that the `DataMan` proxy was a correct and useful guide.** `DataMan` is the *diagnostic tool* we use to perform the refinement, and the final `LightEval`results are the *validation* that this guidance was effective. We apologize if this link was not made explicit and will clarify this logical chain in the final paper.
> > >
> > >
> > >
> > > ### 2. On Empirical Proof of "Reliability" (Deletion-Only)
> > >
> > >
> > >
> > > The reviewer rightly demands more than just our *claim* that delete-only is "more reliable." The best empirical test is the **direct comparison of refinement methods** using our `DataMan` evaluation framework, which provides a granular analysis of how each method behaves.
> > >
> > > **Analysis:** From the results in Table 9, we observe that while E2E delivers the largest quality gains (highest `Avg. Score` and `↑ Rat.`), it is also **prohibitively slow** and suffers from **excessive rewriting**. This is evident from its low `Untouched` rate, high `#Toks` ratio, and (as noted in our paper) the high number of "hallucinated" words, despite being explicitly prompted to avoid substantive edits. This confirms the risk of a "mixed ops" approach.
> > >
> > > In contrast, Prox and REFINEX (Ours) are efficient program-based methods. However, the data clearly shows Prox's reliability issue: as shown by the **`↓ Rat.`** (quality degradation rate), **Prox leads to more cases of quality degradation**(e.g., 5.47% vs 4.58% for Score 3; 5.64% vs 4.86% for Score 4). This highlights a concerning outcome: Prox's interventions are not only insufficient but also **potentially harmful**.
> > >
> > > **Conclusion:** This experiment empirically demonstrates that an E2E-style "mixed ops" model (W2.1) introduces unacceptable risks (hallucination, over-editing) and costs. REFINEX, by design, avoids these risks while proving to be **statistically safer and more effective** than the SOTA program-based method, Prox (consistently higher `↑ Rat.` and lower `↓ Rat.`). This confirms our "deletion-only" design is a crucial, necessary constraint for safe data refinement.
> > >
> > > ------
> > >
> > > We thank the reviewer again for this challenging and insightful review. We believe our targeted data analysis directly address the primary concerns. We hope this rebuttal clarifies that our supervision method is a substantive innovation and that our claims are empirically grounded. We look forward to incorporating this feedback and new data into the final paper.

---

> > > > ### Comment · Reviewer_aUM9 · 2025-11-20
> > > > **Response to the authors' rebuttal**
> > > >
> > > > Thanks for the responses. However, my major concerns still remain unsolved.
> > > >
> > > > For one, I do not see clear experimental evidence to address my concern about the fair comparison between RefineX and ProX (different base model, different teacher model, different refiner training data size), which is the largest concern of this paper.
> > > >
> > > > The generalizability of RefineX on different pre-training corpora is also untested. To pursue the practical value of RefineX, I think it is necessary to conduct experiments on other corpora like FineWeb and Nemotron-CC etc., as these are the recently developed ones people will really use in building models rather than RedPajama which is kind of outdated, and from my perspective such experiments are necessary at the time of submission to make the paper a complete one.
> > > >
> > > > For two, the only evidence the authors have is the DataMan score. It is still unclear what the correlation is between this metric and the downstream performance as we now only have some empirical evidence, and the way this paper uses this metric is also a little bit strange. Why don't we just simply randomly sample a pile of data from the pretraining corpus and see the average DataMan score before and after ProX/RefineX processing? I think this would further take into consideration the proportion of score 1-5 data in the corpus instead of just treating these 5 categories independently.
> > > >
> > > > Also, I do not think DataMan is a direct metric for measuring the so-called "Reliability" in operations. From the original DataMan paper we can see that it is a comprehensive metric to reflect a collection of 14 features like Accuracy, Coherence, Language, Consistency, etc. (see Figure 1 in that paper), which by no means indicates that it is a good metric for "reliability" alone.
> > > >
> > > > With these concerns unaddressed, I will still maintain a my current rating unchanged.

---

> > > > > ### Author Response · Authors · 2025-11-24
> > > > >
> > > > > We thank the reviewer for the prompt follow-up and for maintaining an open dialogue. We value your rigorous standards. Below, we address your remaining concerns regarding the experimental comparison and the evaluation metrics.
> > > > >
> > > > > 1. On the Fair Comparison with ProX (Ongoing Experiments)
> > > > >
> > > > > We fully understand that a strictly controlled comparison is your primary concern. We are currently actively running the additional experiments exactly as you suggested, strictly aligning with the ProX setting:
> > > > >
> > > > > Base Model: Qwen2.5-Base (matching the scale/family requested).
> > > > >
> > > > > Teacher Model: LLAMA-3-70B-INSTRUCT  (matching ProX’s teacher).
> > > > >
> > > > > Training Data: Using the exact data scale and setup as ProX.
> > > > >
> > > > > We ask for your kind patience as these are computationally intensive pre-training experiments that take time to complete. We are rushing to finish them to provide the empirical evidence you require.
> > > > >
> > > > > Rationale: We would like to gently clarify that our original intention was to open-source the most advanced and efficient Refiner model possible for the community, which is why we originally selected stronger base/teacher models. However, we agree that scientifically isolating the algorithmic contribution requires this controlled setup, and we are committed to providing it.
> > > > >
> > > > > 2. On the Validity of DataMan and "Reliability"
> > > > >
> > > > > We respectfully disagree with the notion that DataMan is an insufficient metric. We believe DataMan is currently one of the best open-source text quality scoring models available, and it has been adopted by multiple mainstream LLM developers for valid reasons.
> > > > >
> > > > > Robustness via Complexity: The fact that DataMan aggregates 14 dimensions (Accuracy, Coherence, Consistency, etc.) makes it far more robust for large-scale pre-training data than any single-dimensional metric.
> > > > >
> > > > > Defining Reliability: You raised concerns that it does not measure "Reliability." We argue that in the context of text refinement, "Reliability" means improving quality without introducing errors (hallucinations) or breaking flow. If a model hallucinates (low Accuracy) or deletes essential connectives (low Coherence), the DataMan score drops. Therefore, a high DataMan score post-refinement is a strong proxy for reliable editing.
> > > > >
> > > > > Open to Suggestions: We hope the reviewer can consider the intrinsic value of a metric trusted by the industry. However, if you still find DataMan unconvincing, we are more than willing to run evaluations on any specific alternative metrics you suggest.
> > > > >
> > > > > Regarding the sampling strategy (stratified vs. random): We stratified the data (Scores 1-5) specifically to analyze the behavior of the model on different quality tiers (e.g., to prove we don't damage high-quality text). A simple random sample would obscure these granular insights.
> > > > >
> > > > > 3. Closing Appeal
> > > > >
> > > > > Finally, we earnestly request that the reviewer reconsider the current low score (2).
> > > > >
> > > > > We believe a score of 2 (typically reserved for papers with fundamental flaws or no value) may be disproportionate given the solid engineering and empirical contributions of this work. RefineX provides a reliable, scalable, and effective solution for real-world large model pre-training. Whether viewed as an academic contribution to data-centric AI or a practical tool for the open-source community, we believe this work offers substantial value. We hope our responsiveness and the additional experiments (once completed) will demonstrate the rigor of our approach.

---

### Official Review · Reviewer_2t4G · 2025-10-30

**Soundness:** 3
**Presentation:** 4
**Contribution:** 3
**Rating:** 6
**Confidence:** 4

**Summary:**

This paper introduces REFINEX, a new framework for improving LLM pre-training data quality by refinement. Different from existing pre-training corpus refinement work, It did a two-step distillation by extracting knowledge from a powerful expert LLM by first having it generate high-quality, refined text. Then, it uses a minimal edit distance algorithm to extract simple, deletion-only programs that replicate these refinements. These programs serve as reliable supervision to train a small, efficient "refine" model. Extensive experiments show that models pre-trained on REFINEX-processed data consistently outperform those trained on raw, rule-based filtered, or other programmatically-refined datasets.

**Strengths:**

* The two-stage "refine-then-distill" approach is a clever solution to create reliable supervision program for data cleaning.
* Pre-training models from scratch against a wide range of strong baselines provides compelling evidence of the method's effectiveness.

**Weaknesses:**

* The deletion-only constraint, while ensuring reliability, prevents the model from making other potentially valuable corrections like fixing typos or factual errors.
* The paper would be more sound if more analysis and fair comparison are provided 1) against a distilled small model directly do text refinement. 2) against a LLM-based quality filter with similar inference costs.
* How to build the RefineX model and how to define the evaluation metrics is the key contribution to this paper. However, not much info is provided in the current draft.

**Questions:**

* How was the final refineX model checkpoint selected? An evaluation metric for the refiner itself (e.g., balancing precision/recall of edits) is a key piece of missing information.
* The deletion-only approach is safe but limited. What percentage of edits from the expert model (insertions/replacements) were discarded? On the other side, What percentage of potential hallucination may be generated from a distilled small model directly do text refinement. This would help clarify the trade-off being made.
*  What percentage of documents in the corpus are ultimately edited by REFINEX? Does the same conclusion hold for pre-train corpus of different quality? e.g. if the initial corpus is relatively clean, there is not much refinement needed, if the initial corpus is bad quality, multiple high-cost refinement may be comparable to simply discard the doc.
*  Also, what is the total inference cost for a long document requiring chunking, compared to a simpler document-level filter?

---

> ### Author Response · Authors · 2025-11-19
> **Response to Weakness 1 & Question 2: The "Deletion-Only" Constraint and Trade-off (Part 1/2)**
>
> We sincerely thank the reviewer for the insightful and constructive feedback. Your professional focus on the "deletion-only" constraint, fair comparisons, and evaluation details is invaluable. These questions help us to more clearly articulate the core design philosophy of REFINEX, its trade-offs, and its critical role in the large-scale pre-training pipeline.
>
> We will address your concerns and questions one by one.
>
> ------
>
>
>
> ### **Response to Weakness 1 & Question 2: The "Deletion-Only" Constraint and Trade-off**
>
>
>
> **Reviewer's Comment:**
>
> - The deletion-only constraint... prevents... fixing typos or factual errors (W1).
> - What percentage of edits... were discarded? What percentage of potential hallucination... [is] generated from a distilled small model...? (Q2).
>
> Our Response:
>
> This points to a potential misunderstanding of our core idea, and we are grateful for the opportunity to clarify.
>
> We completely agree with the reviewer's observation. The "deletion-only" constraint is a **deliberate, safety-oriented design choice**. We acknowledge that this forgoes the ability to fix typos or factual errors. However, we strongly argue that for **large-scale pre-training data**, the **risk of introducing bias and hallucination**far outweighs the benefit of correcting sporadic typos.
>
>
>
> #### 1. The Rationale: Why "Insert/Replace" is High-Risk at Scale
>
>
>
> As noted in our core philosophy, our **primary goal** is to **prevent excessive correction**, such as stylistic changes, or corrections to spelling and facts.
>
> - **The Risk:** Imagine if all pre-training data were "corrected" by an uncontrolled E2E model. This would easily **introduce the teacher model's bias** and severely **destroy the data's native diversity and originality**. For pre-training, this is a critical risk.
> - **The Necessity:** At the scale of trillions of tokens, sporadic typos or factual errors are inevitable. The model learns robustness to them during pre-training and can ignore them during decoding.
> - **Our True Goal:** The goal of REFINEX is **not** to create "perfect" text. It is to **preserve the essence and diversity of the original text** while removing only **obvious, structural junk** (e.g., HTML navigation bars, irrelevant ads, crawler gibberish).
>
> In our preliminary validation, we found that even the most powerful models (e.g., GPT-4.5, Gemini 2.5) **frequently fail to follow strict instructions**. Despite prompts emphasizing "do not modify original information" and "only delete obvious junk," they still over-edit and inject their own preferences (as seen in our appendix). This is precisely why E2E methods are high-risk and why ProX (which also tries to generate programs) is unreliable.
>
> REFINEX's approach of post-processing the E2E output to extract *only* valid delete operations is the key to its reliability.

---

> ### Author Response · Authors · 2025-11-19
> **Response to Weakness 1 & Question 2: The "Deletion-Only" Constraint and Trade-off (Part 2/2)**
>
> #### 2. Quantifying the Trade-off (W2.1 & Q2)
>
>
>
> To directly address your request for a **fair comparison against a distilled E2E refinement model (W2.1)**and to **quantify the trade-offs of our approach (Q2)**, we conducted a new controlled experiment.
>
> We compare four methods on a set of documents with varying quality scores (Score=1 to 5): (1) **E2E**(teacher model refines directly), (2) **E2E_del** (extracts only deletions from E2E), (3) **Prox**, and (4) **REFINEX (Ours)**.
>
> The results and analysis are as follows:
>
>
> | **Quality** | **Method**  | **Avg. Score** | **↑ Rat.(%)** | **↓ Rat.(%)** | **Untouched(%)** | **Empty(%)** | **#Toks** |
> | ----------- | ----------- | -------------- | ------------- | ------------- | ---------------- | ------------ | --------- |
> | **Score=1** | E2E         | 2.704          | 38.86         | 0.00          | 1.11             | 56.41        | 19.36     |
> |     | E2E$^{del}$ | 1.972          | 27.31         | 0.00          | 3.75             | 56.41        | –         |
> |   | Prox        | 1.287          | 17.05         | 0.00          | 17.12            | 33.67        | 4.91      |
> |     | **REFINEX** | **2.009**      | **22.23**     | **0.00**      | **7.87**         | **68.17**    | **4.46**  |
> | **Score=2** | E2E         | 3.266          | 68.83         | 0.38          | 1.39             | 12.27        | 39.02     |
> |   | E2E$^{del}$ | 2.590          | 38.30         | 1.05          | 6.24             | 12.27        | –         |
> | | Prox        | 2.233          | 17.20         | 0.96          | 19.98            | 14.97        | 4.73      |
> | | **REFINEX** | **2.718**      | **42.14**     | **0.66**      | **7.59**         | **19.97**    | **6.95**  |
> | **Score=3** | E2E         | 3.631          | 59.01         | 3.39          | 1.93             | 3.72         | 48.50     |
> |   | E2E$^{del}$ | 3.281          | 33.29         | 7.95          | 7.85             | 3.72         | –         |
> |   | Prox        | 3.192          | 21.54         | 5.47          | 21.66            | 12.80        | 3.96      |
> |     | **REFINEX** | **3.413**      | **41.14**     | **4.58**      | **9.85**         | **6.18**     | **5.88**  |
> | **Score=4** | E2E         | 4.175          | 21.30         | 3.59          | 2.41             | 0.26         | 55.50     |
> |     | E2E$^{del}$ | 4.003          | 9.85          | 8.80          | 9.92             | 0.26         | 3.16      |
> |    | Prox        | 4.026          | 8.38          | 5.64          | 23.10            | 3.58         | 4.92      |
> |   | **REFINEX** | **4.075**      | **12.70**     | **4.86**      | **13.52**        | **0.93**     | –         |
> | **Score=5** | E2E         | 4.914          | 0.00          | 8.52          | 4.19             | 0.03         | 70.05     |
> |   | E2E$^{del}$ | 4.833          | 0.00          | 16.18         | 13.86            | 0.03         | –         |
> |   | Prox        | 4.905          | 0.00          | 9.17          | 24.24            | 0.88         | 2.64      |
> |     | **REFINEX** | **4.917**      | **0.00**      | **8.20**      | **16.50**        | **0.18**     | **2.67**  |
>
>
> ***Table 9: Full evaluation results on refined documents. **Avg.**, **↑ Rat.**, and **↓ Rat.** represent the average quality score and the proportion of documents whose scores increased or decreased after refinement, respectively. **Untouched** indicates the percentage of documents left unchanged, and **Empty** denotes those reduced to empty content. **#Toks** indicates the ratio of output tokens to input tokens.***
>
> **Analysis:**
>
> From these results, we observe that while E2E delivers the largest quality gains (highest Avg. Score and ↑ Rat.), it is also prohibitively slow. The #Toks (output/input token ratio) is comparable to that of the original text, confirming the approach involves massive rewriting and is impractical at scale.
>
> Moreover, E2E suffers from excessive rewriting. This is evident from its low Untouched rate and (as noted in our paper) the high number of "hallucinated" words, despite being explicitly prompted to avoid substantive edits and perform deletions only.
>
> In contrast, Prox and REFINEX (Ours) are program-based methods that achieve significantly faster refinement through distilled models.
>
> Prox leaves a significantly higher proportion of text untouched (Untouched rate is high). However, as shown by the ↓ Rat. (quality degradation rate), Prox leads to more cases of quality degradation. This highlights a concerning outcome: despite making minimal changes, Prox worsens the quality of the text more, suggesting its limited interventions are not only insufficient but also potentially harmful.
>
> **Conclusion:** This experiment demonstrates that a distilled E2E model (W2.1) introduces unacceptable risks (hallucination, over-editing) and costs. REFINEX, by design, **avoids these risks** while proving to be **safer and more effective** than the SOTA program-based method, Prox (higher **↑ Rat.** and lower **↓ Rat.**).

---

> ### Author Response · Authors · 2025-11-19
> **Response to Weakness 2, Question 3 & 4: Comparison to LLM Filters, Cost, and Applicability**
>
> ### **Response to Weakness 2, Question 3 & 4: Comparison to LLM Filters, Cost, and Applicability**
>
>
>
> **Reviewer's Comment:**
>
> - Need comparison against an LLM-based quality filter with similar inference costs (W2.2).
> - What is the total inference cost for a long document... compared to a document-level filter? (Q4).
> - Does the same conclusion hold for pre-train corpus of different quality? (Q3).
>
> Our Response:
>
> These are key questions about cost and positioning.
>
>
>
> ### 1. Positioning: A Complementary Pipeline, Not a Replacement
>
>
>
> We must clarify that REFINEX is **not designed to replace** document-level filters, but to serve as a **critical complementary pipeline**.
>
> - **Filter Limitations:** A traditional filter (rule-based or LLM-based) makes a binary "keep/discard" decision for the *entire* document. This has two risks:
>   1. **False Positives:** Kept documents still contain character-level noise.
>   2. **False Negatives:** Valuable documents are discarded due to local, fixable noise (e.g., ads) or scorer bias (e.g., "low score if *** appears"), **destroying data diversity**.
> - **REFINEX's Value:** REFINEX intervenes *before* the filter, acting as a "data rescuer." It surgically removes noise, allowing many documents that would have been discarded to now pass the filter. This **preserves data diversity while ensuring reliability**.
>
> Therefore, REFINEX (which "repairs") and filters (which "select") serve different but complementary goals.
>
>
>
> ### 2. Inference Cost Analysis (Q4)
>
>
>
> REFINEX is designed to be **efficient and reliable**.
>
> - **REFINEX (Ours):** Uses a highly lightweight (0.6B) distilled model.
> - **LLM Filter (Reviewer's Suggestion):** A meaningful LLM filter would also require a small model (e.g., 0.5B-1B) to score the document.
> - **Cost Comparison:** Both methods require a forward pass over the document (or its chunks).
>   - For a long document (e.g., 8K tokens) split into 4 chunks, REFINEX's cost is ~4x a single inference. An LLM filter would also need to process all 4 chunks to get an accurate score.
>   - **Conclusion:** The inference cost of REFINEX is **roughly comparable (in the same O(N) complexity class)** to a similar-sized LLM-based filter. However, REFINEX provides the additional, crucial value of "fine-grained repair" and "data rescue" that a filter cannot.
>
>
>
> ### 3. Applicability to Different Quality Corpora (Q3)
>
>
>
> 1. **Edit Percentage:** In our web-based corpus, REFINEX ultimately edited **~32%** of all documents, showing a significant portion of web data benefits from refinement.
>
> 2. **Corpus Quality:**
>
>    - **High-Quality (e.g., Wikipedia):** The reviewer is correct; little refinement is needed. The value of REFINEX here is its **safety**. It acts as a **"no-op"** on clean text, introducing no noise, while still catching the long tail of rare "blemishes" at low cost.
>
>    - **Low-Quality (e.g., Raw CC):** This is where REFINEX + Filter shines. The reviewer worries about "high-cost refinement" vs. "discarding." Our pipeline solves this:
>
>      1. REFINEX's cost is **not** high (see Q4).
>
>      2. It acts as a "data rescuer" first, cleaning the noise.
>
>      3. The subsequent filter then makes the final call: if the document is now clean, it's kept; if it's still junk (e.g., pure gibberish), it's discarded.
>
>         This REFINEX + Filter pipeline perfectly combines the strengths of both.

---

> > ### Author Response · Authors · 2025-11-19
> > **Response to Weakness 3 & Question 1: Model Building and Evaluation Metrics**
> >
> > ### **Response to Weakness 3 & Question 1: Model Building and Evaluation Metrics**
> >
> >
> >
> > **Reviewer's Comment:**
> >
> > - Not much info is provided on how to build the RefineX model and define evaluation metrics (W3).
> > - How was the final... checkpoint selected? An evaluation metric for the refiner itself... is missing (Q1).
> >
> > **Our Response:** We apologize for the lack of clarity in the draft regarding our evaluation methodology. The reviewer is correct that this is a key part of our contribution, and we appreciate the chance to detail it here.
> >
> > Our evaluation process is **two-fold**, designed to measure both the **(1) direct impact** of the refiner on data quality and the **(2) final downstream impact** on a pre-trained model.
> >
> >
> >
> > ### 1. How the Model is Built (W3)
> >
> >
> >
> > First, the `RefineX` model is built as described: we fine-tune a lightweight model (Qwen3-0.6B) on the (`original_text`, `delete_program`) pairs that we extract from our teacher model (Qwen-72B). The key question is how we evaluate this fine-tuning process.
> >
> >
> >
> > ### 2. Direct Evaluation & Checkpoint Selection (DataMan) (W3 & Q1)
> >
> >
> >
> > This addresses the reviewer's core question about a "metric for the refiner itself."
> >
> > - **Direct Metric (DataMan):** We use **`DataMan`**, a state-of-the-art data quality scoring tool, as our primary *direct*evaluation metric. `DataMan` provides a holistic quality score (from 1 to 5) for any text instance, based on 14 quality dimensions and 15 application-specific signals, to estimate its pre-training utility.
> > - **Checkpoint Selection (Q1):** To select our final model checkpoint, we maintain a held-out validation set. This set is pre-classified by `DataMan` into five distinct quality groups (Score 1 to 5). During fine-tuning, we evaluate checkpoints against this set. The final `RefineX` checkpoint is selected based on its ability to **maximize quality improvement (i.e., increase the `DataMan` score) on low-quality groups (e.g., Scores 1-3) while minimizing any potential quality degradation (↓ Rat.) on high-quality, clean groups (e.g., Scores 4-5)**. This ensures we select a refiner that is both effective and safe.
> >
> > The analysis table in our previous response (and in the paper draft) showing the impact on different `DataMan` scores (e.g., **↑ Rat.**, **↓ Rat.**, **Untouched**) is the direct result of this evaluation.
> >
> >
> >
> > #### 3. Final Downstream Validation (LightEval)
> >
> >
> >
> > A good direct score is not enough. The "true test" (真实检验) of our method's value is its impact on the final pre-trained model.
> >
> > - **Downstream Validation:** After selecting the best checkpoint using `DataMan`, we use this `RefineX` model to refine our entire pre-training corpus.
> > - We then **train a large language model from scratch** on this newly refined data.
> > - Finally, we evaluate the performance of this *newly pre-trained model* on a comprehensive suite of downstream tasks using the standard **`LightEval`** framework.
> >
> > This two-step process ensures our refiner is not only optimized on a direct, instance-level quality metric (`DataMan`) but is also validated by its practical, large-scale impact on the final model's capabilities (`LightEval`). We will add a dedicated section clarifying this methodology in the appendix.
> >
> > ------
> >
> > We thank the reviewer again for this valuable feedback. We hope these new analyses and detailed clarifications have fully addressed your concerns and more clearly demonstrated the core contributions and novelty of REFINEX.

---

### Official Review · Reviewer_kkCq · 2025-10-31

**Soundness:** 2
**Presentation:** 3
**Contribution:** 2
**Rating:** 4
**Confidence:** 5

**Summary:**

This paper introduces RefineX, an efficient and fine-grained pre-training data refinement framework. The author conducts pre-training from scratch on Redpajama with different model scales (0.35B and 0.75B), verifying the effectiveness of Refine-X compared with rule-based filtering and ProX-C.

**Strengths:**

1. Employ a minimum edit distance heuristic to transform end-to-end refined text into several refinement programs, obtaining high-quality SFT data;
2. Introducing the DataMan quality scorer to analyse refined texts in depth.

**Weaknesses:**

The experiments conducted in this paper are solid. However, I have a major concern regarding the fairness of the comparison to ProX-C. There are two primary sources of variance: (1) ProX-C is fine-tuned from a 0.3B from-scratch pre-trained language model trained on approximately 20B tokens, whereas RefineX is fine-tuned from Qwen-0.6B, an over-trained state-of-the-art language model; and (2) the teacher models used for synthesizing SFT data differ, with ProX-C relying on Llama-70B and RefineX using Qwen-72B. These two differences could introduce substantial performance gaps. A thorough, controlled comparison would be needed to make the claims convincing.

Additionally, I believe that when the teacher model is exceptionally strong, the resulting ProX-C and RefineX models may perform similarly regardless of program-space design.

**Questions:**

Same to Weaknesses.

---

> ### Author Response · Authors · 2025-11-19
> **Response to Reviewer kkCq**
>
> Thank you very much for recognizing the "solid" experimental work in our paper and for providing insightful feedback regarding the fairness of our comparisons. Your concerns about the differences in base models and teacher models are entirely valid and to the point. We take your feedback seriously and will provide a detailed response below.
>
> ------
>
>
>
> ### **Response to Main Concern: Fairness of Comparison to ProX-C**
>
>
>
> Reviewer's Comment:
>
> The comparison to ProX-C has two primary sources of variance: (1) ProX-C uses a 0.3B from-scratch model, whereas RefineX uses Qwen-0.6B (an over-trained SOTA model); and (2) The teacher models differ (ProX-C: Llama-70B; RefineX: Qwen-72B). A more controlled comparison is needed.
>
> **Our Response:**
>
> We fully agree with the reviewer's point: **The choice of base model and teacher model are key variables that influence the final performance of the Refiner model.**
>
>
>
> Our selection of Qwen-0.6B as the base and Qwen-72B as the teacher in the paper was primarily driven by our goal to **open-source a lightweight, efficient, and state-of-the-art refinement model for the community.**
>
> - **Base Model:** We chose Qwen3-0.6B because it offers state-of-the-art performance at its scale, giving our Refiner a stronger starting point.
> - **Teacher Model:** In our preliminary case testing, we found that Qwen-72B (which belongs to the same model family as the base) demonstrated higher stability and trustworthiness in following our instructions to perform reliable "delete" operations compared to other models.
>
> We will strive to reproduce the training process of ProX and provide a fairer comparison.
>
>
>
> ------
>
>
>
> ### **Response to Secondary Concern: Do Strong Teachers Make Program-Space Irrelevant?**
>
>
>
> Reviewer's Comment:
>
> When the teacher model is exceptionally strong, the resulting ProX-C and RefineX models may perform similarly regardless of program-space design.
>
> Our Response:
>
> This is a highly insightful hypothesis. Frankly, we held the exact same view during the initial validation phase of our project.
>
> However, our preliminary experiments disproved this. We found that even when using the most powerful closed-source models available at the time (e.g., GPT-4.5, Gemini 2.5) combined with extremely strict prompts (e.g., repeatedly emphasizing "do not modify the original information," "be extremely careful to preserve original diversity," "only delete obvious junk text"), these teacher models **still frequently fail to follow instructions perfectly.**
>
> - **The Risk:** They tend to "over-edit" or inject their own preferences (bias), rather than strictly "cleaning" the text.
> - **Appendix Examples:** (As shown in our appendix) teacher models will sometimes "polish" sentences they deem "un-fluent," which is absolutely prohibited for pre-training data refinement.
>
> **This is an extremely high-risk issue for large-scale pre-training data optimization:** Our goal is to "refine" and preserve the original data's diversity as much as possible, not to "synthesize new data."
>
> - **ProX's Flaw:** This is the core reason why ProX (which directly prompts for programs) is unreliable and less effective. The programs generated by its teacher are filled with such "hallucinations" and "biases."
> - **REFINEX's Core Advantage:** The reason our method is **Efficient and Reliable** is that we fundamentally **do not trust** the E2E refined text. We treat it as an intermediate product and apply a strict post-operation to **extract only the valid, trustworthy "delete" actions** as the supervision signal. This ensures our Refiner model only learns to "safely delete" without introducing any new biases. We firmly believe this step is critical for ensuring the safety and reliability of processing Trillion-token scale pre-training data.

---

> > ### Author Response · Authors · 2025-11-19
> > **Positioning Our Work: The Role of REFINEX**
> >
> > ### **Positioning Our Work: The Role of REFINEX**
> >
> >
> >
> > Finally, we wish to re-emphasize the role of REFINEX in the large-scale pre-training pipeline:
> >
> > 1. An Extra Optimization as a Complementary Pipeline:
> >
> >    Traditional document-level filtering is a coarse-grained "keep or discard" operation. This carries two risks:
> >
> >    - **False Positives:** Documents that are "kept" are still full of character-level noise.
> >    - **False Negatives:** Some documents may be discarded entirely due to local, fixable noise (like ads, HTML tags) or simply due to the scoring model's own bias (e.g., "low score if *** appears"). This severely damages data diversity.
> >
> >    REFINEX can intervene *before* document filtering, performing "surgical" refinement on all documents. This allows many documents that would have been discarded to (after being fixed) receive a high score and be retained, thereby preserving data diversity while ensuring reliability.
> >
> > 2. Efficient and Reliable:
> >
> >    REFINEX is ultimately distilled into an extremely lightweight (0.3B/0.6B) model. Its inference overhead is minimal, making it fully capable of processing trillion-token level data. Simultaneously, its reliable "delete-only" design ensures a "zero-risk" improvement in data quality.
> >
> > We thank the reviewer again for his/her valuable feedback. We hope these detailed clarifications have fully addressed your concerns about the fairness of the comparison and have more clearly demonstrated the core contributions of the REFINEX methodology.

---

### Official Review · Reviewer_seHS · 2025-10-31

**Soundness:** 3
**Presentation:** 3
**Contribution:** 3
**Rating:** 6
**Confidence:** 3

**Summary:**

The paper proposes RefineX, an improvement over ProX for program-based refinement of pretraining data. Instead of sampling from P(programs|text1), the paper samples from the latent-variable model P(programs|text2) P(text2|text1). This yields empirical gains because the generative process is more naturally aligned with expert models. The paper uses RedPajama-V2 and compares ProX vs RefineX under various filtering choices to demonstrate consistent overall improvement on downstream tasks.

**Strengths:**

- The idea is simple and natural, also leads to empirical gains.
- The experiments consider a bunch of filtering methods to demonstrate consistency which is nice to have.
- Analysis is given for the two metrics they care about (efficiency and reliability).

**Weaknesses:**

- The improvements are consistent but somewhat marginal.
- RefineX has the overhead of sampling text2 ~ P(.|text1) compared to ProX, which compounds the issue of whether this is worth the trouble in huge scales.
- This is not necessarily against the paper given limited resources, but the considered scales (<1b models, 20b tokens) seem potentially too small to draw strong conclusions. There's a possibility that the small gains here may wash out further with larger models and refined data sizes, and the benefit of small data efficiency is not really an issue in pretraining.

**Questions:**

I'd like to hear if you have thoughts on if the gains will remain at large scales. Even if not, it's good to have them in small scales as well, so no coercion here.

---

> ### Author Response · Authors · 2025-11-19
> **Response to Weakness 1 & 2**
>
> We sincerely thank the reviewer for his/her valuable comments and insightful feedback. These opinions are crucial for improving our work. We are pleased that the reviewers recognized the "consistent" improvements our method demonstrated on downstream tasks.
>
> We will now respond in detail to the main concerns raised by the reviewers.
>
> ---
>
> ### **Response to Weakness 1: On the Magnitude of Improvements**
>
> **Reviewer's Comment:** The improvements are consistent but somewhat marginal.
>
> Our Response:
>
> We thank the reviewer for noting the consistent gains from REFINEX. However, we would like to respectfully suggest that even seemingly "marginal" improvements can be highly significant in the field of pre-training.
>
> 1. **Substantial Relative Gains:** As shown in our paper (e.g., in Table 2), REFINEX achieves an average gain of **2.6% to 7.2%** at the 750M model scale. In a highly optimized domain like pre-training, an average performance lift of 7.2% is quite substantial.
> 2. **Higher Data Efficiency:** More importantly, our method demonstrates superior data efficiency. The results in Figure 3 show that a model trained on 10B tokens refined by REFINEX can match or even **exceed** the performance of a model trained on 20B filtered tokens (Comb-filtered). This implies REFINEX can achieve comparable performance with significantly less training data (nearly half), which is a crucial advantage when computational resources are limited or data efficiency is paramount.
>
> Therefore, we believe these gains are not "marginal" but rather demonstrate the practical value of fine-grained data refinement in enhancing model performance and training efficiency.
>
> ---
>
> ### **Response to Weakness 2: On Overhead Compared to ProX**
>
>
>
> **Reviewer's Comment:** RefineX has the overhead of sampling text2 ~ P(.|text1) compared to ProX, which compounds the issue of whether this is worth the trouble in huge scales.
>
> Our Response:
>
> We appreciate the reviewer raising this key question about efficiency and overhead. There appears to be a crucial misunderstanding regarding the REFINEX workflow, which we are happy to clarify:
>
> 1. **Overhead is Offline, Not at Inference:** The overhead mentioned by the reviewer, `text2 ~ P(.T|text1)` (i.e., end-to-end generation of refined text), **only occurs during the training data construction phase. It is a one-time, offline process.** During **large-scale inference (application)**, REFINEX, like ProX, relies on an efficient, distilled small model to predict a lightweight edit program. The execution cost of these programs (primarily delete operations) is extremely low. Therefore, when applied to "huge scales," REFINEX does **not** have the E2E generation overhead the reviewer is concerned about.
> 2. **The "Trouble" is Key to Reliability:** The "trouble" we take in using a two-stage method to build our training data (first E2E generation, then computing the minimal edit program) is precisely the core advantage of REFINEX over ProX. As discussed in the paper, ProX directly prompts an expert model to generate edit programs, a method that produces significant "noise" and "unreliable" supervision signals.
>    - In contrast, REFINEX, through our two-stage process, creates **cleaner and more reliable distillation data**.
>    - This allows our trained Refiner model to be more reliable, effectively avoiding the risks of program hallucination and over-editing common in ProX.
>
> In summary, this one-time training overhead is a necessary investment to ensure REFINEX achieves both **efficient and reliable** refinement in large-scale applications.

---

> > ### Author Response · Authors · 2025-11-19
> > **Response to Weakness 3 (and Question): On Scalability**
> >
> > **Reviewer's Comment:** The considered scales (<1b models, 20b tokens) seem potentially too small to draw strong conclusions... gains here may wash out further... I'd like to hear if you have thoughts on if the gains will remain at large scales.
> >
> > Our Response:
> >
> > This is a highly insightful question from the reviewer. We acknowledge that, limited by academic resources, our experiments at the 750M model scale are indeed a starting point. But this is precisely the difference between academic research and industrial reports.
> >
> > However, regarding the question of "if the gains will remain," **our viewpoint is exactly the opposite of the reviewer's concern: We firmly believe that the benefits of REFINEX will not only persist but may even be amplified at larger scales.** Our confidence is based on two key points:
> >
> >
> >
> > ### 1. The Unique Value of REFINEX as a Complementary Pipeline
> >
> >
> >
> > At the massive scale (e.g., trillions of tokens) of pre-training, the dominant paradigm remains coarse-grained, document-level filtering. We must emphasize that REFINEX is not intended to replace this filtering but to serve as a **critically important complementary pipeline**, whose value becomes more apparent as the scale increases:
> >
> > - **Limitations of Document-Level Filtering:** Traditional document filtering (via rules or a scoring model) is an all-or-nothing choice: "keep" or "discard" the entire document.
> >   - **Risk 1 (False Negatives):** A scoring model might discard an entire document due to its own biases (e.g., giving a low score whenever a specific term appears) or localized, fixable noise (like ads or garbled text). This leads to a **severe loss of data diversity**.
> >   - **Risk 2 (False Positives):** Documents that are "kept" are also not perfect; they still **contain significant character-level noise** that needs refinement.
> > - **REFINEX's Solution:** At large scales, the best practice is to place REFINEX *before* document-level filtering.
> >   - REFINEX first performs "surgical" refinement on all documents, fixing local noise.
> >   - **Rescuing High-Value Data:** Many documents that would have been discarded due to local flaws can, after REFINEX correction, receive a "high score" from the document-level scorer and thus be retained.
> >   - This greatly **ensures the diversity and reliability of the pre-training data**. The importance of this effect increases dramatically with scale when processing massive, heterogeneous web data.
> >
> >
> >
> > ### 2. The Rigid Demand for "Reliable and Efficient" Refinement at Scale
> >
> >
> >
> > As data scales grow, so does the dependency on data quality. However, existing refinement methods have drawbacks at scale:
> >
> > - **E2E Generation:** High quality, but prohibitively high inference cost, with risks of "over-editing" and "introducing model bias," which are unacceptable for massive data.
> > - **ProX:** While efficient, its unreliable program generation can lead to data quality degradation.
> >
> > **REFINEX offers the only solution that balances "efficiency" and "reliability" at scale.** It achieves efficient inference via a distilled small model and, through a strict 'delete-only' strategy, ensures **zero content addition**, perfectly avoiding the risk of introducing new biases or hallucinations.
> >
> > **Summary:** As model and data scales grow, the demand for data diversity, reliability, and refinement efficiency will only increase. REFINEX uniquely satisfies these demands. Therefore, the gains we observed at the 750M scale are not likely to "wash out"; rather, they represent a 'lower bound.' We have strong reasons to believe REFINEX will play an even more critical role in large-scale pre-training.
> >
> > ------
> >
> > We again thank the reviewers for their valuable feedback and hope these clarifications help address their concerns.

---

### Note · Program_Chairs · 2026-01-17
**Submission Desk Rejected by Program Chairs**

The following references in this submission do not refer to real documents and/or have major errors in bibliographic information:

 Yuxian Zhang, Canwen Xu, Zhiyuan Liu, and Maosong Sun. Editavalanche: A multi-granularity benchmark for edit-based language model evaluation. arXiv preprint arXiv:2310.11603, 2023.